# Wave-ice interactions in the neXtSIM sea-ice model

Timothy D. Williams[1], Pierre Rampal[1], and Sylvain Bouillon[1]

[1]Nansen Environmental and Remote Sensing Center, Thormøhlensgate 47, N5006, Bergen, Norway and the Bjerknes Center for Climate Research, Bergen, Norway.

*Correspondence to:* T. D. Williams (timothy.williams@nersc.no)

**Abstract.** In this paper we describe a waves-in-ice model which calculates ice breakage and the wave radiation stress (WRS) that is coupled to the new sea-ice model neXtSIM, which is based on the Elasto-Brittle (EB) rheology. We highlight some numerical issues involved in the coupling, and investigate the impact of the WRS, and of modifying the EB rheology to lower the stiffness of the ice in the area where the ice has broken up (the marginal ice zone, or MIZ).

In experiments in the absence of wind, we find that wind waves can produce noticeable movement of the ice edge in loose ice (concentration around 70%) — up to 36 km, depending on the material parameters of the ice that are used, and the dynamical model used for the broken ice. The ice edge position is unaffected by the WRS if the initial concentration is higher ($\gtrsim 0.9$). Swell waves (monochromatic waves with low frequency) do not affect the ice edge location (even for loose ice), as they are attenuated much less than the higher frequency components of a wind wave spectrum, and so consequently produce a much lower WRS (by about an order of magnitude at least).

In the presence of wind, we find that the wind stress dominates the WRS, which while large near the ice edge, decays exponentially away from it. This is in contrast to the wind stress which is applied over a much larger ice area. In this case (when wind is present) the dynamical model for the MIZ has more impact than the WRS, although that effect too is relatively modest. When the stiffness in the MIZ is lowered due to ice breakage, we find that on-ice winds produce more compression in the MIZ than in the pack, while off-ice winds can cause the MIZ to be separated from the pack ice.

## 1 Introduction

Wave-ice interactions have received a great deal of attention in recent years (e.g. Dumont et al., 2011; Kohout et al., 2014; Ardhuin et al., 2016, 2017), with progress in both modelling and measuring (particularly via Synthetic Aperture Radar imagery, or SAR) of waves in ice. To a large extent, this is due to climate change, with a series of record lows in both minimum and maximum Arctic sea-ice extents in the last decade (e.g. Meier, 2017).

Specifically, large parts of the Arctic are becoming and are expected to become even more accessible for resource exploitation and shipping in the summer, whereas 10 years ago they weren't (e.g. Stephenson et al., 2011). Associated with this low sea-ice extent is an increased open-water fetch available for wave generation which means there are potentially more large wave events in the Arctic in summer (e.g. in the Beaufort Sea in summer 2012; Thomson and Rogers, 2014). As well as being dangerous for shipping in themselves, large waves also increase the amount of ice breakage in the marginal ice zone (MIZ), creating an extra hazard as small floes could potentially be thrown onto a ship deck for example.

Closely connected to waves in ice, but with other controlling factors apart from waves, is the concept of floe size distribution (FSD e.g. Toyota et al., 2011; Herman, 2010). This can influence both the dynamics and thermodynamics of the ice, ocean and atmosphere in the MIZ. For example, it affects sea-ice rheology (Herman, 2012; Feltham, 2005) and can increase wind/ocean drag and consequently increase the stresses applied to the ice. It can also enhance lateral melting in summer (Horvat et al., 2016; Steele, 1992). Horvat et al. (2016) showed that increased horizontal salinity gradients at the floe edges produced eddies which allowed warm water to travel under the ice floes and enhance the melting from the edges. This was true even for large floes ($\sim 1$ km), when the lateral-to-horizontal-surface-area ratio is quite small. (Previously, this ratio was used to compute results which indicated lateral melting was unimportant for floes larger than $\sim 100$ m; Steele, 1992.) Models for full numerical FSD's (Zhang et al., 2016), where a histogram of floe size bins can evolve in time, as well as joint ice thickness and floe size distributions have been proposed (Horvat and Tziperman, 2015). In the latter model, each thickness category can have its own FSD. More parametric approaches have also been used (Dumont et al., 2011; Williams et al., 2013a; Bennetts et al., 2017).

On the sea-ice modelling side, there has been a lot of progress in making sea-ice dynamics more realistic, especially in the Arctic pack. Rampal et al. (2016) presented a validation of the neXt-generation Sea Ice Model neXtSIM, looking at sea-ice area and extent, sea-ice drift, and the spatial scaling of sea-ice deformation derived from SAR (see also Bouillon and Rampal, 2015b). The dynamical core of neXtSIM is the EB sea-ice rheology, which is a thin elastic plate model with stresses constrained by a Mohr-Coulomb failure envelope. If stresses become too large and leave this envelope in a grid cell, the ice stiffness inside that cell is reduced (in practice a parameter called the damage is increased) in order to bring the stresses back onto the failure envelope (Rampal et al., 2016, for more details). When one cell is highly damaged, the likelihood for the surrounding cells to also become damaged is increased, leading to the rapid (i.e. after a few sea-ice-model time steps) emergence of very localised lines of damaged cells where sea ice can deform almost freely. These lines of concentrated damage can accommodate large deformation (i.e., opening, ridging and shearing) in a way that is similar to the so-called linear kinematic features that are observed from satellites (Kwok, 2001).

In this paper we demonstrate the coupling of a waves-in-ice model (WIM) to neXtSIM in an idealised domain. The physical effects included in the coupling are the break-up of ice by waves, the wave radiation stress (WRS), and an additional (optional) feedback to the sea-ice model where the ice stiffness is reduced where the ice is broken (in the MIZ). We conduct experiments with waves by themselves to see the impact of the WRS on the ice edge location, and also with wind to see the relative importance of the wind stress and the WRS. In addition, we do some simulations to see the particular effects of the rheological change.

We also highlight some general numerical issues involved with coupling wave models and sea-ice models on different grids. In addition, we do some theoretical reformulations of the WIM to put the ice break-up model in the context of Mohr-Coulomb failure, and do some sensitivity tests of the sensitivity of the MIZ width to the Young's modulus in particular, as well as the small-scale "cohesion" parameter in the WIM breaking model. Its response to the Young's modulus was previously uninvestigated.

## 2  Sea-ice model

### 2.1  Evolution equations

The ice is modelled as a thin elastic plate (e.g. Fung, 1965, §16.8) with constitutive relation

$$\boldsymbol{\sigma} = \mathbf{C}(Y_*, \nu)\boldsymbol{\varepsilon}, \tag{1}$$

or in full:

$$
\begin{pmatrix} \sigma_{11} \\ \sigma_{22} \\ \sigma_{12} \end{pmatrix}
= \frac{Y_*}{1-\nu^2}
\begin{pmatrix} 1 & \nu & 0 \\ \nu & 1 & 0 \\ 0 & 0 & (1-\nu)/2 \end{pmatrix}
\begin{pmatrix} \varepsilon_{11} \\ \varepsilon_{22} \\ 2\varepsilon_{12} \end{pmatrix}, \tag{2}
$$

where $\sigma_{ij}$ and $\varepsilon_{ij}$ $(i,j = 1,2)$ are respectively the stress and strain tensors, $\nu$ is Poisson's ratio and $Y_*$ is the effective Young's modulus (depending on the concentration $c$ and the damage $d$), given by

$$Y_*(c,d) = Y_0(1-d)\mathrm{e}^{-C(1-c)}, \tag{3}$$

where $C$ is the compactness parameter, and $Y_0$ is the Young's modulus of fully-compacted, undamaged ice.

The momentum balance equation we will use is the following:

$$\rho_{\mathrm{i}} h \frac{D\mathbf{u}}{Dt} = \nabla \cdot (\boldsymbol{\sigma} h) - \nabla P + \tau_{\mathrm{a}} + \tau_{\mathrm{o}} + \tau_{\mathrm{w,i}}; \tag{4}$$

here $\rho_{\mathrm{i}}$, $h$ and $\mathbf{u}$ are the density, actual thickness, velocity and internal stress tensor of the ice (respectively), $\nabla = (\partial_x, \partial_y)^{\mathrm{T}}$ is the horizontal gradient, and $\tau_{\mathrm{o}}$ and $\tau_{\mathrm{a}}$ are the applied stresses by the ocean and the atmosphere (respectively). These latter stresses come from quadratic drag laws. Note that we neglect the Coriolis force and the gravitational force due to the slope of the ocean surface because of our idealised domain. Also appearing in (4) are the wave radiation stress (WRS), $\tau_{\mathrm{w,i}}$, and the term involving $P$, which is a strictly positive pressure that provides a resistance to compaction and ridging (i.e., it is only activated when the divergence $\nabla \cdot \mathbf{u} < 0$):

$$P = \max\left\{0, -\frac{P_* h^2 \mathrm{e}^{-C(1-c)} \nabla \cdot \mathbf{u}}{|\nabla \cdot \mathbf{u}| + \dot{\varepsilon}_{\min}}\right\}, \tag{5}$$

where $P_*$ is the pressure parameter, and $\dot{\varepsilon}_{\min} = (0.01/86400)\,\mathrm{s}^{-1}$ is the minimum divergence rate. If the ice becomes very damaged, and loses its stiffness, this term prevents the ice from piling up and becoming too thick. As a default, we use the standard value of $P_* = 12\,\mathrm{kPa}$, as suggested by Thorndike et al. (1975), but we will test the sensitivity of our results to $C$ (see §5.3). $C = 20$ is commonly used in the standard sea-ice models using a Viscous Plastic (VP) rheology, so the pressure drops by a factor of about 55 when the open water fraction increases from 0 to 20%. So, for example, increasing $C$ to 40 means the open water fraction only needs to be 10% for the pressure to reduce by 55.

We also have equations for evolution of any conserved quantity $\phi$:

$$\frac{\mathrm{D}\phi}{\mathrm{D}t} = -\phi(\nabla \cdot \mathbf{u}) + S_\phi; \tag{6}$$

$\phi$ could be concentration ($c$, also requiring $c \leq 1$), volume ($ch$) or variables relating to the damage (retrieved from $(1-d)^{-1}$). The terms $S_\phi$ are thermodynamic source/sink terms which are switched off for this paper, since the simulations are in an idealised setting and run for short durations. In an Eulerian frame of reference,

$$\frac{\mathrm{D}\phi}{\mathrm{D}t} = \frac{\partial \phi}{\partial t} + \mathbf{u} \cdot \nabla \phi, \tag{7}$$

but since we work in a Lagrangian frame the relationship is simply $\mathrm{D}\phi/\mathrm{D}t = \mathrm{d}\phi/\mathrm{d}t$. The $-\phi(\nabla \cdot \mathbf{u})$ term represents the conserved quantity decreasing if the divergence is positive e.g. if a triangle in the finite element mesh increased in area then $\phi$ should drop in that triangle.

Like Williams et al. (2013a), we will parameterise the floe size distribution in terms of the maximum floe size, $D_{\mathrm{max}}$ (see §3.3), which we wish to advect like a tracer: $\mathrm{D}(D_{\mathrm{max}})/\mathrm{D}t = 0$. In the Lagrangian framework, advection is usually exact, unless a local remeshing is required. This happens if the triangles of the mesh become too deformed, and requires (local) interpolation of the advected variable. Details on the remeshing procedure in the neXtSIM model can be found in Rampal et al. (2016). Additional (global) interpolation is required to obtain $D_{\mathrm{max}}$ on the fixed grid of the WIM (see §4). We found that transporting and interpolating $D_{\mathrm{max}}$ itself led to some errors, which were reduced by transporting an auxiliary variable $N_{\mathrm{floes}} = c/D_{\mathrm{max}}^2$ according to

$$\frac{\mathrm{D}}{\mathrm{D}t}\left(\log(N_{\mathrm{floes}})\right) = \frac{\mathrm{D}}{\mathrm{D}t}\left(\log(c)\right), \tag{8}$$

or to progress from (neXtSIM) time step $n$ to $n+1$, $N_{\mathrm{floes}}$ should change according to $N_{\mathrm{floes}}^{(n+1)} = c^{(n+1)} N_{\mathrm{floes}}^{(n)}/c^{(n)}$, and being interpolated when either regridding or communication with the WIM is required.

The evolution of stress and damage from time step $n$ to $n+1$ is done via an intermediate stress calculation:

$$\boldsymbol{\sigma}' = \boldsymbol{\sigma}^{(n)} + \mathbf{C}(c,d)\dot{\boldsymbol{\varepsilon}}\Delta t, \tag{9a}$$

$$\boldsymbol{\sigma}^{(n+1)} = \Psi \boldsymbol{\sigma}', \tag{9b}$$

$$d^{(n+1)} = 1 - \Psi(1-d^{(n)}) + \Phi_d \Delta t, \tag{9c}$$

where $\Phi_d$ is a thermodynamic source term (again not used here), while $\Psi$ ($0 < \Psi \leq 1$) is a factor determined from the position of the stress vector relative to the Mohr-Coulomb failure envelope, described in section 2.3. There is no continuous version of (9) since fracturing is an extremely rapid process, well below our typical time step $\Delta t$.

## 2.2 Uncoupled neXtSIM simulation

Since the damage variable $d$ is probably unfamiliar to most readers, we include here an example simulation illustrating its main role in the EB rheology. Figure 1 shows four fields after a 2-day simulation. The wind stress plotted is calculated from the quadratic drag law

$$\boldsymbol{\tau}_{\mathrm{a}} = \rho_{\mathrm{a}} C_{\mathrm{d,a}} |\mathbf{u}_{\mathrm{a}} - \mathbf{u}|(\mathbf{u}_{\mathrm{a}} - \mathbf{u}), \tag{10}$$

where $\rho_a = 1.3\,\mathrm{kg\,m^{-3}}$ and $\mathbf{u}_a$ are the density and 10-m-velocity of the air, while $C_{d,a} = 7.6 \times 10^{-3}$ is the drag coefficient of the wind on the ice. The gradient in the wind stress comes from the differences in relative velocity. We have plotted this stress as a reference for when we discuss the WRS.

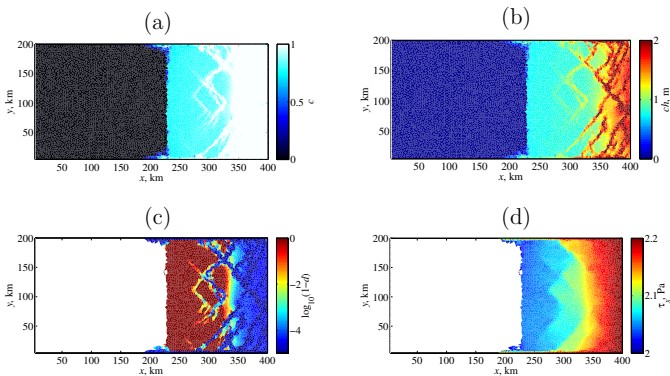

**Figure 1.** Results after forcing from uniform, steady wind (with speed $14.9\,\mathrm{m\,s^{-1}}$, from the left) has been applied for 48 h. Initially, constant ice conditions were applied ($c = 0.7$, $h = 1\,\mathrm{m}$, $d = 0$) were applied to the right of the ice edge, which corresponded to approximately $x = 124\,\mathrm{km}$. The upper, lower and right hand boundaries are closed. Fields plotted are (a) concentration, (b) effective thickness, (c) the damage (with blue being more damaged and red less), and (d) the $x$-component of the wind stress $\boldsymbol{\tau}_a$. There are no wave interactions considered, C=40 and $\tau_0^L = 4\,\mathrm{kPa}$.

Initially, the concentration was relatively low, so the internal stress was also low (see the formulae for $Y_*$ and $P$ in equa-
5    tions 3, 5), meaning the ice was almost in free drift, being compressed against the right hand boundary. As the concentration increased, the internal stress increased causing it to fail (increase $d$) in localised regions. Comparing the damage with the concentration and thickness, it can be seen that the regions of high compression and thickening correspond to the regions where the damage is highest. This is the usual role (without waves) of the damage — to produce localised deformation and features such as thicker regions (under shearing or convergent conditions, such as in the current simulation) and leads (under shearing or
10    divergent conditions). We note here that the initial combination of $c = 0.7$ (loose ice) with no damage is not inconsistent since the damage only increases if the concentration is high, although the reasons it is usually initialised to zero are: (i) for simplicity and (ii) since it is not an observable variable. It then evolves with the other variables in response to the applied forcings.

## 2.3 Mohr-Coulomb failure

Let $\sigma_1$ and $\sigma_2$ be the principal stresses, with compressions corresponding to positive stresses. Then a stress state is within the Mohr-Coulomb failure envelope if the conditions

$$\sigma_2 \leq \sigma_c + q\sigma_1, \quad \sigma_1 \leq \sigma_c + q\sigma_2, \tag{11a}$$

$$\sigma_{N,min} \leq \sigma_N \equiv \frac{1}{2}(\sigma_1 + \sigma_2) \leq \sigma_{N,max}, \tag{11b}$$

are satisfied (Schulson et al., 2006; Dansereau et al., 2016; Rampal et al., 2016), where

$$\sigma_c = \frac{2\tau_0}{\sqrt{\mu^2 + 1} - \mu}, \quad q = \left(\sqrt{\mu^2 + 1} + \mu\right)^2, \quad \sigma_{N,min} = -\frac{5\sigma_c}{6(q-1)}, \quad \sigma_{N,max} = \frac{75}{4}\tau_0,$$

and $\tau_0$ is the cohesion, and $\mu$ is the internal friction coefficient. See Figure 2(a) for some example envelopes ($\tau_0 = 629$ and 989 kPa). The lines $\sigma_2 = \sigma_c + q\sigma_1$ and $\sigma_1 = \sigma_c + q\sigma_2$ in the space of principal stresses (Schulson et al., 2006; Dansereau et al., 2016) correspond to the lines $|\tau| - \mu\sigma_N = \tau_0$, so the material fails when the applied shear force $|\tau|$ reaches the sum of the frictional force inside the material ($\mu\sigma_N$) and the cohesion of the material ($\tau_0$). Now

$$|\tau| = \frac{1}{2}(\sigma_1 - \sigma_2)(\sin(2\vartheta) + \mu\cos(2\vartheta)) \leq \frac{1}{2}(\sigma_1 - \sigma_2)\sqrt{\mu^2 + 1} \tag{12}$$

(Schulson et al., 2006), where $\vartheta$ is the angle between the maximum principal stress (taken as the most compressive stress), $\sigma_1$, and the failure plane. This reaches its maximum value when $\tan(2\vartheta) = 1/\mu$, so if $\mu = 0.7$, the failure plane is oriented at about $27.5°$ from the direction of $\sigma_1$. Equation (12) also lets us derive the expressions for $q$ and $\sigma_c$.

The conditions (11b) are less certain since there are fewer measurements in pure tension or compression. In particular, extending the Coulomb branches into the third quadrant in principal stress space (see Figure 2 of Dansereau et al., 2016, who instead apply tensile failure criteria $\sigma_1, \sigma_2 \geq -\sigma_c/q$) could be seen as theoretically suspect (since there should be no friction under tension), but the observations of Weiss et al. (2007; see Figure 2) seem to support this approach. In practice, using $\sigma_N \geq \sigma_{N,min}$ or $\sigma_1, \sigma_2 \geq -\sigma_c/q$ was found to make little difference to large-scale simulations. Similarly, $\sigma_{N,max}$ is set large enough that it is not reached in simulations, which is reasonable since few examples of large biaxial compressive stresses have been observed (Weiss et al., 2007). Note that Dansereau et al. (2016) chose not to close the failure envelope at all for this same reason.

Returning to (9), if $\boldsymbol{\sigma}'$ is outside the envelope it is scaled back onto the nearest branch of the envelope by setting $\boldsymbol{\sigma}^{(n+1)} = \Psi\boldsymbol{\sigma}'$, where $\Psi < 1$. This ensures that the stress always remains within the envelope, but the damage $d$ is increased if this happens. Otherwise, if $\boldsymbol{\sigma}'$ is inside the envelope, $\Psi = 1$ and the damage is unchanged.

## 2.4 Scaling of the Mohr-Coulomb envelope

Mohr-Coulomb envelopes have been observed on many different scales in rock mechanics, and has also been seen in ice. The parameter $\mu$ controls the orientation of fractures that form, while the cohesion sets the sizes of the stresses which cause any fractures, and so is more influential.

This property should scale as $\tau_0 \propto L_c^{-1/2}$, where $L_c$ is the size of the defects, or "stress concentrators" (Weiss, 2013, §4.2). Put in another way,

$$\frac{\tau_{0,0}}{\tau_{0,1}} = \sqrt{\frac{L_{c,1}}{L_{c,0}}}, \tag{13}$$

where the additional indices 0 or 1 correspond to different scales on which fracture is occurring. Table 1 shows the Mohr-Coulomb parameters, and the estimated defect sizes, which have been fitted to various time series of stress measurements.

| Measurement type | $\tau_0$ | $\mu$ | $L_c$ | Reference |
|---|---|---|---|---|
| Lab | 1.1 MPa | 0.92 | 1.3 mm | Schulson et al. (2006) |
| *In situ* | 40 kPa | 0.7 | 1 m | Weiss et al. (2007) |
| Reference simulation | 4 kPa | 0.7 | 100 m | Bouillon and Rampal (2015a) |
| *In situ* | 1 kPa | 0.7 | 1.6 km | Weiss et al. (2007) |

**Table 1.** Cohesion values, internal friction coefficient from measured Mohr-Coulomb failure envelopes. Also given are approximate defect sizes deduced from these envelopes, using the scaling law (13). (These defect sizes, or sizes of stress concentrators, are only meant to give an idea of the relative sizes compared to those corresponding to the second cohesion value which is approximated to be around 1 m which is of the same order as the ice thickness. The first defect size is of the same order as the grain size — the grains measured in the sample were columns of diameter 3.9 mm and length 1 cm.) For some additional context, we also give the value used in the reference simulation of Bouillon and Rampal (2015a). This large-scale cohesion is in contrast to our small-scale cohesion ($L_c \sim 1$ cm), which we use to determine if single ice floes will fracture due to wave flexure.

Note that these values do not necessarily correspond to the breaking stress of ice since the measurements are not exactly taken at the point of fracture. The lab measurement (uni-axial compression test) should be closer since we know the ice did actually break and the scale of the measurement; the *in-situ* measurements are certainly underestimations since the ice did not break, and in fact the value of 1 kPa was derived from a 3-day subset of the time series which was bounded by the envelope with cohesion 40 kPa. That is, the lower *in-situ* value corresponds to more remote fracturing, or fracturing over a larger scale.

In their presentation of the dynamical core of the neXtSIM model (using a resolution of approximately 10 km), Bouillon and Rampal (2015a) found that the model was quite sensitive to the cohesion value when varied between 0.5 kPa and 8 kPa. However, the results for $\tau_0^L = 8$ kPa (the superscript 'L' here indicates it is the large-scale cohesion, as opposed to the small-scale one discussed below) and $\tau_0^L = 4$ kPa were similar. In the follow-up paper to the aforementioned one, Rampal et al. (2016) used $\tau_0^L = 8$ kPa, or $L_c \approx 25$ m. This gave good agreement with the deformation-scaling statistics.

In the simulations done in this paper we will use a model resolution of 4 km, so we will test a range of cohesions from 4–13 kPa to be somewhat consistent with the above choice. Also, we will discuss the ice breakage by waves (below in §3.4.1) in terms of Mohr-Coulomb failure, and define an additional small-scale cohesion $\tau_0^S$ and defect scale $L_c$ for the breaking criterion we settle on in section 3.4.2.

## 3 Waves-in-ice model

### 3.1 Attenuation

The amount of attenuation that waves in ice experience is the main factor in determining the amount of momentum transferred to the ice. However, definitive confirmation of any particular physical models for this is still lacking. Meylan et al. (2014) came up with an empirical formula fitted to Antarctic attenuation from the experiments reported by Kohout et al. (2014). Ardhuin et al. (2016) compared the creep model of Wadhams (1973) (also see Tolman *et al.*, 2016, §2.4) with drifting buoy data from within the ice, with some success in the timing of the peaks in wave heights. Other theoretical models that have been used are a viscoelastic attenuation model (Wang and Shen, 2010), and "localisation" predicted by 1D multiple scattering models (Kohout and Meylan, 2008; Bennetts and Squire, 2012). In the wave scattering context, localisation refers to how these models predict exponential decay of waves as they travel into the ice. Or in other words, the wave energy is localised in the vicinity of the ice edge.

Doble and Bidlot (2013) used the model of Kohout and Meylan (2008) in Antarctic simulations using WAM, while Williams et al. (2013a) used a theoretical result from Bennetts and Squire (2012) to investigate break-up by waves. Tolman *et al.* (2016, §2.4) give a full summary of waves-in-ice parameterisations implemented in Wavewatch III.

Our attenuation model is essentially model B from Williams et al. (2013a), slightly modified to allow Young's modulus to be varied. It has a scattering component determined from the expected number of floes per unit length, and a dissipative component coming from the drag model of Robinson and Palmer (1990)

$$\alpha_{\text{scat}} = \frac{\alpha c}{\langle D \rangle}, \quad \alpha_{\text{dis}} = 2c\beta; \tag{14}$$

here, $\alpha$ is the scattering per floe, while $\beta$ is the imaginary part of the wave number satisfying the dispersion relation of Robinson and Palmer (1990), calculated using the method of Williams et al. (2013a, Appendix A) with drag coefficient $\Gamma = 13\,\text{Pa}\,\text{s}\,\text{m}^{-1}$.

As stated above, the choice of attenuation model is crucial in determining the wave radiation stress, yet physical mechanisms are still relatively uncertain. However, we can still calculate the response of the ice to waves attenuated by our model, and make conclusions which should still hold for similar ranges of the WRS.

### 3.2 Energy transport

A general formulation for wave energy transport is

$$\frac{\partial E}{\partial t} + \mathbf{C}_{\text{g}} \cdot \nabla E = S_{\text{in}} + S_{\text{nl}} + S_{\text{ice}}, \tag{15a}$$

$$\frac{1}{c_{\text{g}}} S_{\text{ice}}(\mathbf{x}, t; \omega, \theta) = (\mathcal{L}_{\text{scat}} - \alpha_{\text{dis}}) E(\mathbf{x}, t; \omega, \theta), \tag{15b}$$

$$\mathcal{L}_{\text{scat}} E = -\alpha_{\text{scat}} E + \int_{0}^{2\pi} K(\theta - \theta') E(\mathbf{x}, t; \omega, \theta') \mathrm{d}\theta'. \tag{15c}$$

where $\mathbf{C}_{\mathrm{g}} = c_{\mathrm{g}}(\cos\theta, \sin\theta)^T$ is the group velocity vector, $c_{\mathrm{g}} = d\omega/dk$, $\omega$ is the radial frequency, $k$ is the wavenumber, and $E$ is the spectral density function (SDF) of the variance of the wave elevation $\eta$:

$$\langle \eta^2 \rangle = m_0, \quad m_n = \int\limits_0^\infty \int\limits_0^{2\pi} E(\mathbf{x}, t; \omega, \theta) \omega^n \, \mathrm{d}\theta \, \mathrm{d}\omega \quad (n = 0, 1, 2, \ldots); \tag{16}$$

the SDF of the time-averaged energy is $E' = \rho_{\mathrm{w}} g E$, where $\rho_{\mathrm{w}}$ is the water density and $g$ the acceleration due to gravity.

We neglect the terms $S_{\mathrm{in}}$ and $S_{\mathrm{nl}}$, which represent wind generation and non-linear energy transfer between frequencies and directions (respectively). The term $S_{\mathrm{nl}}$ moves energy from high frequencies to lower ones, and becomes more significant if $E$ is larger. For example, Kohout et al. (2014) described a storm event off Antarctica (with approximate latitude 61°S and longitude 125°E) where the significant wave height was measured to decay linearly with distance into the ice, whereas it decayed exponentially during calmer periods. Li et al. (2015) attributed this to the effect of $S_{\mathrm{nl}}$, and the fact that lower frequencies are attenuated less than higher ones. Thus we need to remember that our results could change (e.g. waves could induce ice breakage further from the edge) if our wave forcing becomes very large. In particular, the WRS may also persist further than predicted with our linear model — however, it would also have a smaller size since the longer waves are attenuated less.

The scattering kernel $K$ distributes energy from the incident wave among the other directions and is discussed further in the next section. Various authors (e.g. Perrie and Hu, 1996; Masson and LeBlond, 1989) have used the solution for a rigid circular floating disc to deduce an expression for $K$; Meylan et al. (1997) extended this to make the disc elastic, and this solution was also used by Zhao and Shen (2016); Ardhuin et al. (2016) used the simpler kernel $K = \alpha_{\mathrm{scat}}/(2\pi)$ to distribute the incident energy uniformly in all directions. However, due to the fact that these models conserve energy, i.e.

$$\int\limits_0^{2\pi} \mathcal{L}_{\mathrm{scat}} E \, \mathrm{d}\theta = 0, \quad \text{or} \quad \alpha_{\mathrm{scat}} = \int\limits_0^{2\pi} K(\theta - \theta') \mathrm{d}\theta \quad \text{for } 0 \le \theta' \le 2\pi, \tag{17}$$

the operator $\mathcal{L}_{\mathrm{scat}}$ has some zero eigenvalues. (This is most easily seen by considering the discretised version of (17) — i.e. considering only a finite number of directions — which would state that all the columns of the matrix representing $\mathcal{L}_{\mathrm{scat}}$ add to zero. Thus the rows are linearly dependant and the matrix will have at least one zero eigenvalue.) This usually means that the solution $E$ of (15) will usually not decay exponentially into the ice (in the absence of dissipation). (This decay depends on the eigenvector(s) corresponding to the zero eigenvalue, of course, but in general they are such that $E$ does not decay into the ice.) As a result, the results of Ardhuin et al. (2016) which included scattering in this way were quite unrepresentative of phase-resolving multiple-scattering models such as those of Kohout and Meylan (2008) and Bennetts and Squire (2012). Consequently, we will use $K = 0$ and not conserve energy, since we think that it is preferable to preserve the localisation predicted by the scattering models.

### 3.3 Floe size distribution

We use a parametric form of the FSD. We initially require that $D_{\max} \ge D_{\min}$ and that large floes ($> 200\,\mathrm{m}$) have a uniform floe size distribution — i.e. $p(D|D_{\max} > 200\,\mathrm{m}) = \delta(D - 200\,\mathrm{m})$. This latter assumption is somewhat vestigial but was related to

the fact that wavelengths that do breaking in the ice are usually less than about 400 m. The rest of our approximation is similar to the FSD used by Dumont et al. (2011), which was based on the renormalisation group (RG) approach to the same problem, used by Toyota et al. (2011). However, this formula made the mean floe size a discontinuous function of the maximum floe size, so we have modified it to a continuous (as opposed to discrete) FSD — a power-law-type probability density function $p(D)$ truncated at $D = D_{\mathrm{max}}$, but with the same exponent as before:

$$
p(D|D_{\mathrm{max}} \leq 200\,\mathrm{m}) = \begin{cases} \dfrac{\gamma D_{\mathrm{min}}^{\gamma} D_{\mathrm{max}}^{\gamma}}{D_{\mathrm{max}}^{\gamma} - D_{\mathrm{min}}^{\gamma}} D^{-(1+\gamma)} & \text{for } D_{\mathrm{min}} \leq D \leq D_{\mathrm{max}}, \\ 0 & \text{otherwise} \end{cases} \tag{18}
$$

where $\gamma = 2 + \log f / \log \xi$, $f$ is the fragility in the RG formulation of Toyota et al. (2011), and $\xi^2$ is the number of pieces formed during each successive break-up in the same RG formulation. We use $D_{\mathrm{min}} = 20$ m, $f = 0.9$ and $\xi = 2$, making $\gamma \approx 1.84$.

Results for the MIZ width (not shown) with the RG approach are similar to those with the FSD (18), but the momentum flux is less smooth, which could cause numerical problems. We recognise that both parameterisations are completely arbitrary, and that numerical histograms (e.g. as used by Horvat and Tziperman, 2015) are preferable in terms of being able to let the wave spectrum try to produce the FSD naturally. (They also let other factors influence the FSD more easily). However, the FSD itself is not the focus of this current paper, and these alternative models are quite costly and not trivial to implement, so we do not try them out here.

## 3.4  Ice breakage due to waves

### 3.4.1  Plane strain and Mohr-Coulomb failure

It is instructive to put the situation of ice breakage due to a plane wave in the context of the discussion in §2.3. We also use a thin elastic plate model, so the constitutive relation is similar to equations (1–2): $\boldsymbol{\sigma} = \mathbf{C}(Y, \nu)\boldsymbol{\varepsilon}$, where $Y$ is the Young's modulus for an ice floe. However, for waves we are interested in the stresses that are induced by a vertical displacement $\eta$. The stresses are assumed to be confined to the horizontal plane and varying linearly with the vertical coordinate $z = x_3$ ($z = 0$ is the middle of the plate, and $-\frac{1}{2}h \leq z \leq \frac{1}{2}h$) (Fung, 1965, §16.9). We then have the following results for stresses and strains

$$
\sigma_{3i} = \sigma_{i3} = \sigma_{33} = \varepsilon_{3i} = \varepsilon_{i3} = 0 \qquad\qquad \text{for } i = 1, 2, \tag{19a}
$$

$$
\varepsilon_{ij} = -z\partial_{x_i}\partial_{x_j}\eta, \ \ \varepsilon_{33} = -\nu(\sigma_{11} + \sigma_{22}) = -\frac{\nu}{1-\nu}\sum_{k=1}^{2}\varepsilon_{kk} \qquad\qquad \text{for } i, j = 1, 2, \tag{19b}
$$

where $x_1 = x$ and $x_2 = y$. For a plane wave (travelling in the $x$ direction with amplitude $A$) in a thin elastic plate, $\eta = A\cos(kx - \omega t)$, $\varepsilon_{11} = k^2 z\eta$, $\varepsilon_{22} = \varepsilon_{12} = \sigma_{12} = 0$, and so the only non-trivial stresses are given by

$$
\sigma_1 = \sigma_{11} = \frac{Y\varepsilon_{11}}{1-\nu^2}, \ \ \sigma_2 = \sigma_{22} = \nu\frac{Y\varepsilon_{11}}{1-\nu^2} = \nu\sigma_1, \tag{20}
$$

where $\sigma_1$ and $\sigma_2$ are the principal stresses in the horizontal plane. This meets the upper Mohr-Coulomb branch when

$$\sigma_2 = \nu\sigma_1 = \sigma_c + q\sigma_1, \tag{21a}$$

$$\sigma_1 = \sigma_1^{(\text{tens})} \equiv -\frac{\sigma_c}{q - \nu} = -\frac{(2\tau_0^S)/(q - \nu)}{\sqrt{\mu^2 + 1} - \mu} \approx -1.13\tau_0^S \tag{21b}$$

(if $\mu = 0.7$, it doesn't meet the lower branch, $\sigma_1 = \sigma_c + q\sigma_2$, if $\sigma_N \geq \sigma_{N,\text{min}}$). Note that here the shape of the tip of the failure
envelope makes a difference, since a pure tensile failure criterion would increase the lower limit on $\sigma_1$ to $-\sigma_c/q \approx -1.04\tau_0^S$
(which would be reached at smaller wave amplitudes). However, given the uncertainty about the failure envelope under pure
tension and high compression, and so that our small- and large-scale envelopes have the same shape, we use (11) for wave
failure also.

Figure 2(a) plots the failure envelopes for two values of the cohesion. The figure also shows where the line corresponding to
the stress state for plane waves, $\sigma_2 = \nu\sigma_1$, meets these Mohr-Coulomb envelopes (i.e., when $\sigma_1 = \sigma_1^{(\text{tens})}$).

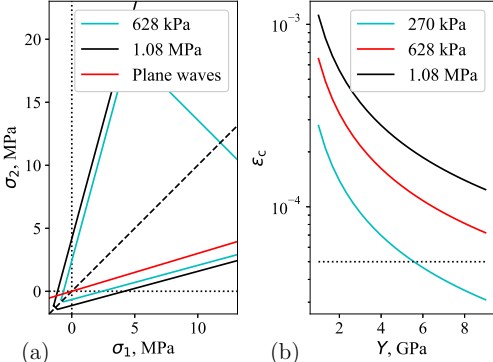

**Figure 2.** (a) Mohr-Coulomb fracture envelope for different values of the cohesion. The red line shows the line $\sigma_2 = \nu\sigma_1$, where $\nu = 0.3$
is Poisson's ratio — this gives the relationship for plane waves in a thin elastic plate. When the ice has thickness $1\,$m, Young's modulus
$5.49\,$GPa, and the wave period is $12\,$s, the red line meets the black one when the wave height is about $60\,$cm. The dashed line shows the
symmetry of the envelopes in the line $\sigma_2 = \sigma_1$. (b) Breaking strain for different values of the cohesion and Young's modulus ($Y$). The dotted
line corresponds to $\varepsilon_c = 5 \times 10^{-5}$.

### 3.4.2 Breaking criterion

The maximum strains are produced when $z = \pm h/2$ (at the upper and lower surfaces of the ice), and so for a plane wave

$$\varepsilon \equiv \max\{\varepsilon_{11}\} = \frac{1}{2}k^2 Ah. \tag{22}$$

Williams et al. (2013a) imposed a strain criterion for breaking, supposing that ice would break if $\varepsilon \geq \varepsilon_c^{\text{est}} = \sigma_f^{\text{est}}/Y$, where $\sigma_f^{\text{est}}$
is the flexural strength estimated from measurements. Timco and Weeks (2010) compiled many measurements for the flexural

strength, fitting the formula

$$10^{-6}\sigma_{\mathrm{f}}^{\mathrm{est}} = 1.76\mathrm{e}^{-5.88\sqrt{v_{\mathrm{b}}}}, \tag{23}$$

where $v_{\mathrm{b}}$ is the brine volume fraction. (It should be noted however, that Karulina et al., 2013, found a different relationship for Barents Sea sea ice.) When considering flexural strength measurements, however, it is useful to remember how they are obtained. In a cantilever situation, an ice beam is subjected to a force $F_{\mathrm{c}}$ at one end until it breaks at the other. The force is then converted to a stress in order to remove the effects of the beam dimensions according to the formula

$$\sigma_{\mathrm{f}}^{\mathrm{est}} = \frac{6F_{\mathrm{c}}L}{h^2 b} \tag{24}$$

(Frederking and Svec, 1985), where $L$ and $b$ are the length and breadth of the beam respectively. (Similar formulae exist for three-and four-point-bending tests.) This conversion assumes that the beam can be modelled as an Euler-Bernoulli beam (e.g. infinitesimally thin and wide). With this model, the only non-zero stress is $\sigma_{11} = Y\varepsilon_{11}$ which would produce Mohr-Coulomb/tensile failure when $\sigma_{11} = -\sigma_{\mathrm{c}}/q$. Hence the flexural strength can be used to estimate the small scale cohesion by

$$\sigma_{\mathrm{f}}^{\mathrm{est}} = \frac{2\tau_0^{\mathrm{S,est}}/q}{\sqrt{\mu^2+1}-\mu} \approx 1.04\tau_0^{\mathrm{S,est}}. \tag{25}$$

The lab measurement of cohesion ($\tau_0^{\mathrm{S}} = 1.1\,\mathrm{MPa}$, Schulson, 2009, also see Table 1) used a sample with $v_{\mathrm{b}} = 0.05$, so $\sigma_{\mathrm{f}}^{\mathrm{est}} \approx 473\,\mathrm{kPa}$ and $\tau_0^{\mathrm{S,est}} \approx 454\,\mathrm{kPa}$ — that is, the estimated failure stress and cohesion are too small, by a factor of approximately 2.42. A similar factor was obtained by Marchenko et al. (2014), who used a full finite element 3D solver (COMSOL) to estimate the stress at the fixed end of a cantilever at the time of breaking, and found it to be approximately $2.6\sigma_{\mathrm{f}}^{\mathrm{est}}$. Now, the results of these simulations depends on the boundary conditions used (e.g. the properties of the spring foundation used; free surface conditions when the ice was partially submerged), and in addition some predictions were not observed (e.g. they predicted the force measured in the tests should increase when the radius of the holes drilled near the beam root increased: Marchenko et al., 2017). However, it gives further indication that $\sigma_{\mathrm{f}}^{\mathrm{est}}$ could definitely be a significant underestimation for the actual breaking stress. If we wanted to be consistent with the lab scale measurement of the cohesion over a range of brine volume fractions, we could propose the relationship $\tau_0^{\mathrm{S}} \approx 2.42\tau_0^{\mathrm{S,est}} \approx 2.33\sigma_{\mathrm{f}}^{\mathrm{est}}$. In practice though, the sensitivity studies are conducted by varying the small scale cohesion directly, and seeing the range of MIZ widths obtained. However, more observations with regard to ice breakage by waves are needed to set a definitive breaking criterion. Some laboratory experiments to this effect are planned to occur in 2018 in the wave/ice tank in Aalto, Finland, as part of the Hydralab+ programme, but field observations would also be very useful.

When we return to our plane wave in an elastic plate, the Mohr-Coulomb criterion is equivalent to the strain criterion

$$\varepsilon \geq \varepsilon_{\mathrm{c}} = \frac{1}{Y}(1-\nu^2)\left|\sigma_1^{(\mathrm{tens})}\right| \approx 1.03\frac{\tau_0^{\mathrm{S}}}{Y}, \tag{26}$$

instead of using $\varepsilon_{\mathrm{c}}^{\mathrm{est}}$. Due to cancellation of unrelated but similar factors this is approximately the same as the breaking strain of Williams et al. (2013a) ($\sigma_{\mathrm{f}}^{\mathrm{est}}/Y$). This ($\varepsilon_{\mathrm{c}}$) is plotted in Fig 2(b) as a function of $Y$. The breaking strain for sea ice (from

beam tests) is typically thought to be about $3 - 10 \times 10^{-5}$ (e.g. Langhorne et al., 1998), but this number contains a lot of assumptions, e.g. about the value of Young's modulus and the stress at the time of breaking (see the discussion below about the flexural strength). In fact, we are not aware of any strain measurements for ice which actually broke. Langhorne et al. (2001) measured strains up to about $3.6 \times 10^{-6}$ in landfast ice which was experiencing incoming waves but which did not break.

Fig 2(b) shows the breaking strains are about the right order ($5 \times 10^{-5}$ is plotted as a dotted line for reference), although higher values of the cohesion combined with lower values of Young's modulus can take them up to $10^{-3}$.

When we have a spectrum of waves, the corresponding quantity to (22) is related to the maximum mean square strain by

$$\frac{\varepsilon^2}{2} \equiv \left\langle \max\{\varepsilon_{11}\}^2 \right\rangle = m_\varepsilon, \quad m_\varepsilon \equiv \frac{h^2}{4} \int\limits_0^\infty \int\limits_0^{2\pi} E(\mathbf{x}, t; \omega, \theta) k^4 \, \mathrm{d}\theta \, \mathrm{d}\omega. \tag{27}$$

If all the wave energy is travelling in one direction (which direction is not relevant since we also do not attempt to consider

an anisotropic wave medium), equation (26) is still equivalent to the Mohr-Coulomb criterion since we still have $\sigma_2 = \nu \sigma_1$. However, we now have a statistical (approximately normal) distribution of strains $\max\{\varepsilon_{11}\}$, instead of a fixed strain amplitude. Thus (26) corresponds to a condition on the probability of $\max\{\varepsilon_{11}\}$ exceeding $\varepsilon_c$

$$\mathbb{P}(\max\{\varepsilon_{11}\} > \varepsilon_c) \geq \mathbb{P}_c, \tag{28}$$

where $\mathbb{P}_c$ is some critical probability. An alternative to (26) could be to choose $\mathbb{P}_c$ another way (e.g. defining it as the ratio of

a breaking time scale to the mean wave period), or else $\mathbb{P}(\max\{\varepsilon_{11}\} > \varepsilon_c)$ could be used directly in a similar formulation to Horvat and Tziperman (2015). However, for now we use (26) so that the criterion agrees with the criterion for a plane wave (e.g. a swell wave).

When the wave energy is not unidirectional, the stresses are no longer distributed on the line $\sigma_2 = \nu \sigma_1$, so the probability condition (28) is no longer equivalent to the Mohr-Coulomb criterion. A simple numerical experiment generating random

waves in an ice sheet and creating an artificial time series (not shown) found that $\mathbb{P}(\max\{\varepsilon_{11}\} > \varepsilon_c)$ was significantly lower than the probability of the stresses leaving the failure envelope (about 45% compared to about 65% in one example). However, for now we will leave this as a *caveat* and attempt a fuller investigation of the Mohr-Coulomb failure in a random sea at a later date.

### 3.4.3   Ice break-up

When (26) is satisfied, we calculate the mean zero crossing frequency from

$$\langle \omega_{02}^2 \rangle = \frac{m_2}{m_0} \tag{29}$$

and convert this to a wavelength $\lambda_{02}$ using the dispersion relation for a thin elastic plate (Williams et al., 2013a, Appendix A). Then $D_{\max}$ is reduced to $\lambda_{02}/2$ (requiring that it stays above $D_{\min} = 20\,\mathrm{m}$, and that it is actually reduced — i.e. it can't increase, since we don't consider thermodynamic effects in this paper).

## 3.5 Momentum loss due to attenuation

Following Phillips (1977, Chapter 3), we first connect the mean energy per unit area (integrated over the entire water column) for a single plane wave to the mean momentum per unit area. The mean kinetic energy density is

$$E_{\mathrm{K}} = \rho_{\mathrm{w}} \left\langle \int_{z_{\mathrm{bot}}}^{\eta} (u_{\mathrm{w}}^2 + v_{\mathrm{w}}^2) \mathrm{d}z \right\rangle \approx \rho_{\mathrm{w}} \int_{z_{\mathrm{bot}}}^{0} \langle u_{\mathrm{w}}^2 + v_{\mathrm{w}}^2 \rangle \mathrm{d}z$$

$$= \rho_{\mathrm{w}} \omega^2 \frac{A^2}{4k} \cosh(kZ) = \rho_{\mathrm{w}} g \frac{A^2}{4}, \tag{30}$$

where $u_{\mathrm{w}}$ and $v_{\mathrm{w}}$ are the horizontal and vertical wave orbital velocities, and $Z$ is the water depth. In a conservative system, the mean potential energy and the mean kinetic energy are equal, so the mean energy density is simply

$$E_{\mathrm{tot}} = 2E_{\mathrm{K}} = \rho_{\mathrm{w}} g \frac{A^2}{2} = \rho_{\mathrm{w}} g \langle \eta^2 \rangle. \tag{31}$$

The mean momentum per unit area is:

$$\mathbf{M} = \left\langle \int_{-Z}^{\eta} (u_{\mathrm{w}}, v_{\mathrm{w}}) \mathrm{d}z \right\rangle = -\rho_{\mathrm{w}} \langle \Phi_{z=\eta} \nabla \eta \rangle \approx -\rho_{\mathrm{w}} \langle \Phi_{z=0} \nabla \eta \rangle$$

$$= \rho_{\mathrm{w}} g \frac{kA^2}{2\omega} (\cos\theta, \sin\theta) = \frac{E_{\mathrm{tot}}}{c_{\mathrm{p}}} (\cos\theta, \sin\theta), \tag{32}$$

where $c_{\mathrm{p}} = \omega/k$ is the phase velocity.

When we consider a complete wave spectrum, then

$$\mathbf{M} = \rho_{\mathrm{w}} g \int_{0}^{\infty} \int_{0}^{2\pi} \frac{1}{c_{\mathrm{p}}} E(\mathbf{x}; \omega, \theta) (\cos\theta, \sin\theta) \mathrm{d}\theta \mathrm{d}\omega, \tag{33}$$

and its flux is

$$\frac{\mathrm{D}}{\mathrm{D}t} \mathbf{M} = \rho_{\mathrm{w}} g \int_{0}^{\infty} \int_{0}^{2\pi} \frac{1}{c_{\mathrm{p}}} \times \frac{\mathrm{D}}{\mathrm{D}t} E(\mathbf{x}; \omega, \theta) (\cos\theta, \sin\theta) \mathrm{d}\theta \mathrm{d}\omega$$

$$= \rho_{\mathrm{w}} g \int_{0}^{\infty} \int_{0}^{2\pi} \frac{1}{c_{\mathrm{p}}} S_{\mathrm{ice}}(\mathbf{x}; \omega, \theta) (\cos\theta, \sin\theta) \mathrm{d}\theta \mathrm{d}\omega. \tag{34}$$

This quantity can then be transferred to the ice, ocean and atmosphere, according to the different attenuation mechanisms, i.e.

$$-\frac{\mathrm{D}}{\mathrm{D}t} \mathbf{M} = \tau_{\mathrm{w,i}} + \tau_{\mathrm{w,o}} + \tau_{\mathrm{w,a}}. \tag{35}$$

For this study we assume that all the momentum goes to the ice — i.e. $\tau_{\mathrm{w,o}} = \tau_{\mathrm{w,a}} = 0$.

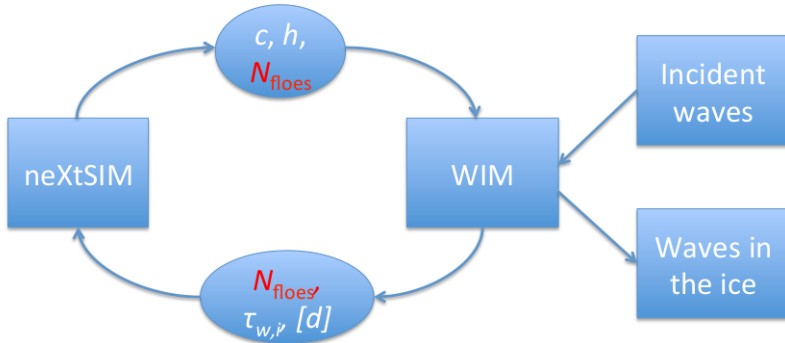

**Figure 3.** Schematic showing the information passed between neXtSIM and the WIM. Note that $N_{\text{floes}}$ is modified by both the WIM and neXtSIM (which use different grids), so must be treated carefully to avoid numerical diffusion. Also input to the WIM are incident wave fields, and it also outputs diagnostic fields of the waves in the ice. Optional: the WIM may also update the damage $d$.

## 4   Coupling to the WIM

Figure 3 shows a schematic diagram of the information passed between the WIM and neXtSIM, as well as external inputs and outputs to and from the WIM. Each time the WIM is called, it takes in the following fields from neXtSIM: $c$, $h$ and $N_{\text{floes}}$. Between calls, these will have changed due to dynamic (advection) and thermodynamic processes (melting, freezing). These are interpolated from the neXtSIM mesh to the WIM grid, and $D_{\max}$ is retrieved from $N_{\text{floes}}$. After the call to the WIM, $N_{\text{floes}}$ is passed back onto the centres of the mesh, and the stresses $\boldsymbol{\tau}_{w,i}$ are interpolated from the grid centres onto the nodes of the mesh, and are used in the solution of the momentum equation. These stresses are kept constant until the next call to the WIM — since the mesh is moving, this requires re-interpolation at each neXtSIM time step.

In an initial more naive implementation of the coupling, $N_{\text{floes}}$ was computed only on the WIM grid, then interpolated back onto the mesh. However, passing this field to and fro between the mesh leads to a large amount of numerical diffusion. To solve this problem, the WIM model takes in the neXtSIM mesh, and each WIM timestep the smoother integrals $m_0$, $m_2$ and $m_\varepsilon$ are interpolated from the grid to the mesh. This allows the breaking calculation to be done on the mesh in parallel to the one on the grid — thus $N_{\text{floes}}$ does not need to be interpolated back to the mesh. This also reduced the diffusion in $N_{\text{floes}}$ significantly. (See Figures 7–8 below.)

The directional wave spectrum is remembered from the previous call, and if necessary can be updated regularly using forcing from an external model, or as in the simulations presented in this paper, using idealised (constant) wave forcing.

We can also change the dynamics of the broken ice. The default, "R0" or rheology 0, does not change the underlying EB rheology. In an alternative, "R1" or rheology 1, we increase the damage parameter $d$ to an arbitrary high value $d_{\text{broken}}$ when the ice is broken by waves. This reduces the internal stress, apart from a pressure term which resists compression, causing the ice velocity to be closer to the free drift velocity.

Alternative continuum approaches to MIZ dynamics are based on the idea of a "granular temperature" (kinetic energy associated with velocity fluctuations relative to the mean flow field). Most recently, Feltham (2005) used a binary collision model to formulate an equation for the granular temperature. Previously, Shen et al. (1986, 1987) had used a similar but simpler approach, where the granular temperature was approximated to be in steady state. This enabled the granular temperature to be found analytically and the constitutive relation to be directly modified without solving any other equations apart from the momentum equations. Shen et al. (1987) compared the granular temperature to field data from the MIZEX campaign of 1983 (Hibler and Leppäranta, 1984), and found it to be correlated, but found that it was an order of magnitude too small. The internal ice stresses were also very low. Feltham (2005) was able to produce some qualitative features such as ice jets in a one-dimensional simulation, but no further comparisons were done. This model is now being introduced into CICE-E (Community ICE code, version E; Rynders et al., 2016).

However, in the field of 3D granular flows, different types of flow regimes have also been observed. For example, the introduction of Guo and Campbell (2016) describes a transition between an inertial collision regime to an inertial non-collisional regime where the stresses follow Bagnold's law (Bagnold, 1954) as the concentration and shear rate increase, and then a further transition to what they call the elastic regime as the concentration and shear rate increase even more. This regime is characterised by the formation of force chains at high concentrations and shear rates, which deform elastically to support the applied stresses.

There have also been a number of direct (discrete) numerical simulations of collections of floes (e.g. Herman, 2013; Rabatel et al., 2015). They have also observed phenomena similar to the force chains mentioned above, where elaborate force contact networks were observed over the full domain of simulation. To summarise, the binary collisional models represent only a small fraction of the types of granular flows observed, so there is much more work required before a complete "MIZ rheology" that could be substituted for our simple modification is ready.

## 5  Results

### 5.1  Note on wave and wind forcing

In our results section we will partly use incident wind wave spectra based on the Bretschneider spectrum:

$$E_{\mathrm{B}}(\omega; H_{\mathrm{s}}, \omega_{\mathrm{p}}) = \frac{5 H_{\mathrm{s}}^2 \omega_{\mathrm{p}}^4}{16 \omega^5} \mathrm{e}^{-(5\omega_{\mathrm{p}}^4)/(4\omega^4)}, \tag{36}$$

where $H_{\mathrm{s}}$ is the significant wave height, $\omega_{\mathrm{p}} = 2\pi/T_{\mathrm{p}}$, and $T_{\mathrm{p}}$ is the peak period.

Since $H_{\mathrm{s}}$ and $T_{\mathrm{p}}$ are not totally independent, to try to make them roughly consistent we will also use a special case of (36), the Pierson-Moskowitz spectrum which was defined as an approximation for fully-developed wind seas:

$$E_{\mathrm{PM}}(\omega; \omega_0) = \frac{a_{\mathrm{PM}} g^2}{\omega^5} \mathrm{e}^{-b_{\mathrm{PM}}(\omega_0/\omega)^4}, \tag{37}$$

where $a_{\mathrm{PM}} = 8.1 \times 10^{-3}$, $b_{\mathrm{PM}} = 0.74$, and $\omega_0 = g/U_{19.5} \approx g/(1.026 U_{10})$. Here $U_{19.5}$ and $U_{10}$ are the wind speeds 19.5 m and 10 m above the sea (respectively) — note that these wind speeds are linked to the incident wave parameters, and we will also try

to keep them consistent when we are presenting coupled WIM-neXtSIM results. The Bretschneider parameters corresponding to the Pierson-Moskowitz parameters are:

$$\omega_{\mathrm{p}} = (4b_{\mathrm{PM}}/5)^{1/4}\omega_0 \approx 0.877\omega_0, \tag{38a}$$

$$H_{\mathrm{s}} = \frac{4g}{\omega_{\mathrm{p}}^2}\sqrt{\frac{a_{\mathrm{PM}}}{5}}. \tag{38b}$$

Our incident wind wave spectra will then combine a Bretschneider frequency spectrum with some directional spreading:

$$E_{\mathrm{inc}}(\omega,\theta; H_{\mathrm{s}},\omega_{\mathrm{p}}) = E_{\mathrm{B}}(\omega; H_{\mathrm{s}},\omega_{\mathrm{p}})D_{\mathrm{inc}}(\theta), \quad D_{\mathrm{inc}}(\theta) = \frac{2}{\pi}\cos^2\theta \times H(|\theta| - \pi/2), \tag{39}$$

where $H$ is the Heaviside step function. (Note that the mean wave direction is zero, ie to the right in our model domain, which can be seen in Figure 1.) We will also look at so-called swell waves, which are not locally generated, generally quite long (wave period greater than about $10\,\mathrm{s}$ or longer), and are monochromatic and mono-directional:

$$E_{\mathrm{swell}}(\omega,\theta; H_{\mathrm{swell}},\omega_{\mathrm{swell}}) = \frac{1}{8}H_{\mathrm{swell}}^2\delta(\omega - \omega_{\mathrm{swell}})\delta(\theta). \tag{40}$$

## 5.2  Sensitivity of MIZ width to Young's modulus and small-scale cohesion

The purpose of this section is to test sensitivity to the Young's modulus and the small-scale cohesion, not necessarily to decide on "correct" values, which are best determined from future observations. The experiments are similar to those of Williams et al. (2013b), although the effect of the Young's modulus was not tested in that paper. This is an interesting parameter since

increasing it makes the ice less compliant and easier to break (ie. a given wave amplitude produces a higher stress in the ice) — potentially increasing the MIZ width — but this also increases the attenuation, which could potentially reduce the MIZ width. The effect of the small-scale cohesion will play a similar role to the breaking strain in that paper.

The Young's modulus is typically somewhere in the range of 1–10 GPa. Williams et al. (2013a) argued for values within the interval 5–7 GPa (depending on the brine volume fraction), proposing that the effective elastic modulus, which includes a

response to primary, recoverable creep, should cause it to drop somewhat from the relationship of Timco and Weeks (2010). However, Marchenko et al. (2013) derived significantly lower values of Young's modulus (about 1.5 GPa) in Svalbard fjord ice. Marchenko et al. (2017) also measured lower values in the Barents Sea, ranging between 1–4 GPa, with no obvious dependance on the brine volume. Therefore, we do some tests of the sensitivity of the MIZ width and the maximum WRS to this parameter.

Figure 4 shows the variation of the MIZ width (panel a) and the maximum WRS (panel b) with peak period for different val-

ues of the Young's modulus. Since increasing the Young's modulus increases the attenuation, the waves lose more momentum and so the maximum radiation stress increases, and this is clearly seen in Figure 4(b). However, Figure 4(a) clearly shows that the MIZ width increases with increasing Young's modulus, so its effect on the breaking criterion clearly dominates its effect on the attenuation. The magnitude of the maximum radiation stress is of the order of 0.1–1 Pa, which is comparable to the wind stress from a $10$–$15\,\mathrm{m\,s^{-1}}$ winds (see Figure 1d). However, while stresses of this size are significant, they are very much

localised around the ice edge as opposed to being applied over large areas (as wind stresses are — see Figure 1(d)).

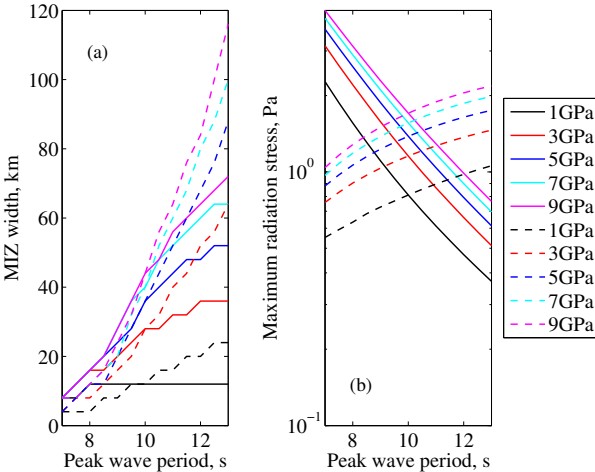

**Figure 4.** Variation of MIZ width (a) and maximum WRS (b) with peak wave period and Young's modulus. Dashed curves: Pierson-Moskowitz spectra are used for the forcing. Solid curves: Bretschneider spectra are used with the significant wave height being 4m. The concentration was 0.7, the thickness was 1 m, and the small-scale cohesion used was 629 kPa. The WIM is not coupled to neXtSIM.

The dashed curves use fully-developed seas (Pierson-Moskowitz spectra), where $H_s$ increases with $T_p$, for wave forcing. Although waves of higher periods are attenuated less, the increasing wave height overcomes this effect and both the MIZ width and maximum radiation stress increases monotonically with peak period.

The solid curves in Figure 4 are created using an incident wave spectrum based on a Bretschneider spectrum with a constant
significant wave height of $H_s = 4$ m. Like with the dashed curves (fully developed seas), larger values of Young's modulus cause the MIZ width to increase monotonically as peak period increases (in the plotted range of periods). However, when $Y = 1$ GPa, as peak period is increased, the MIZ width is initially 8km, then increasing to a maximum of 12 km as the wave frequencies with the most energy are attenuated less, before dropping down to 8 km again as the waves with the most energy, while still being attenuated less strongly, now produce less strain (see equations (22–27)).

This latter result ($Y = 1$ GPa, constant wave height) is similar to results for constant-amplitude swell waves, plotted in Figure 5 — very low periods are attenuated too strongly to do much breaking so the MIZ width is zero; above a certain period the MIZ width increases (with period) to a maximum then drops back down to zero when the induced strain is no longer large enough to cause breakage. For this wave height of 3 m, which is relatively large, but not unrealistic for the usual range of swell periods (ca. 10–20 s), the maximum radiation stress drops from about 0.1 Pa to about 0.01 Pa showing the reduced ability of
swells to produce wave drift in comparison to wind seas.

Figure 6 shows the variation of the MIZ width with the peak period and the small scale cohesion. Unlike the Young's modulus, this parameter does not change the attenuation directly, and so the maximum radiation stress is essentially the same for all values of the cohesion (notwithstanding small differences, mainly due to the different MIZ widths, since the attenuation is higher in the MIZ in our model).

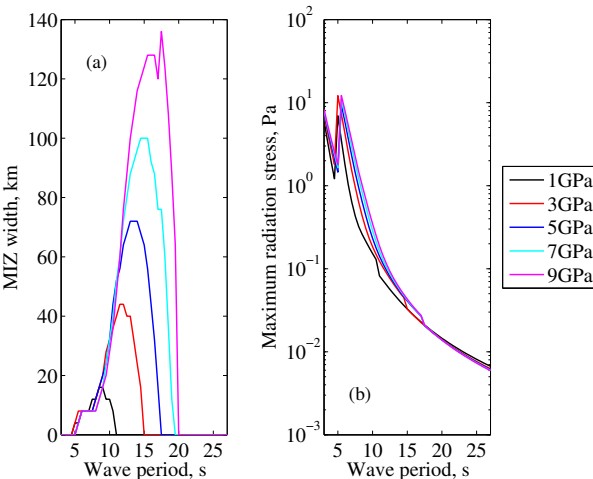

**Figure 5.** Variation of MIZ width (a) and maximum radiation stress (b) with peak wave period and Young's modulus for swells of height 3 m. The concentration was 0.7, the thickness was 1 m, and the small-scale cohesion used was 629 kPa. The WIM is not coupled to neXtSIM.

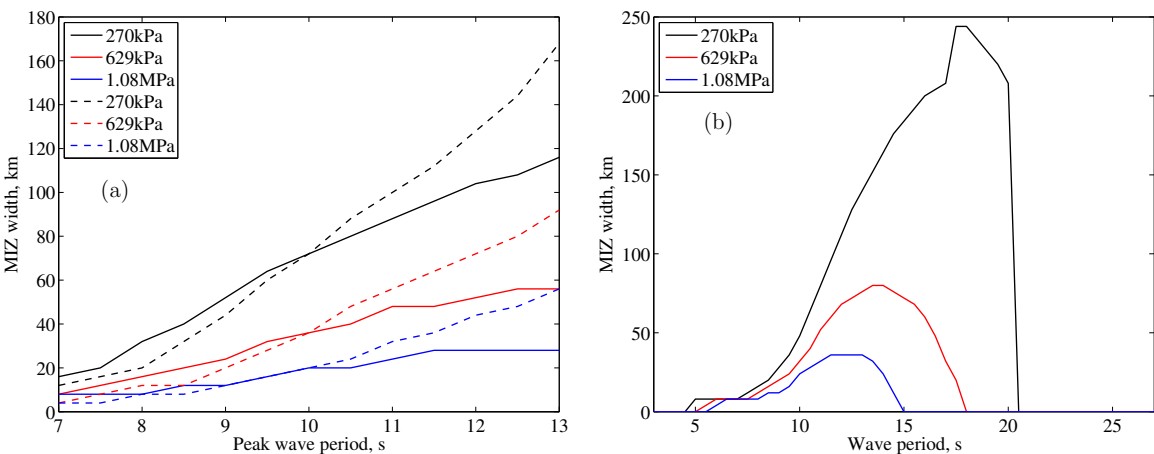

**Figure 6.** Variation of MIZ width with peak wave period and small-scale cohesion for (a) wind seas and (b) swells. (a) Dashed curves: Pierson-Moskowitz spectra are used for the forcing. Solid curves: Bretschneider spectrum are used with the significant wave height being 4m. (b) Swell waves of height 3 m. For both plots, the concentration is 0.7, the thickness is 1 m, and the Young's modulus used was 5.49 GPa. The WIM is not coupled to neXtSIM.

The three values chosen are 270 kPa (approximately the flexural strength when $v_b = 0.1$, 274 kPa), 629 kPa (approximately $2.33 \times 274 = 638$ kPa), and 1.08 MPa (approximately 1.1 MPa, the laboratory value of the cohesion). The results for the MIZ width are significantly different but all are in the correct order of magnitude (a few tens of kilometers). Therefore we will use

$\tau_0 = 629 \, \text{kPa}$ throughout the rest of the paper. We will also use a Young's modulus of $Y_0 = Y_* = 5.49 \, \text{GPa}$ (i.e. the same value in neXtSIM and the WIM).

## 5.3 Coupled waves-in-ice results

Figure 7 shows plots of different fields after a 2-day simulation with neXtSIM coupled to the WIM. There is no wind, only waves arriving from the left (the initial wave state is shown in Figure 7(a)), breaking the ice and pushing it to the right by about 24 km by the end of the 48-h simulation. The initial ice state is the same as in Figure 1, but with the addition of unbroken ice ($D_{\max} = 300 \, \text{m}$ everywhere where $c > 0$), as shown in Figure 7(b). This could correspond to summer ice in the Fram Strait where there can be large floes with large gaps between them (perhaps due to smaller floes melting faster), producing a low concentration.

The resulting MIZ width is about 50 km, which is not unrealistic. Following (39), there is a cos-squared type of directional spreading applied (and 16 directions used) and the upper and lower grid cells, which contain land, act to completely absorb the waves. Therefore, in Figure 7(c), the waves are slightly lower (by about 1 m) near the coast than they are at the centre. In Figure 7(f), the $x$-component of the WRS is plotted — note that while it reaches 1 Pa in the vicinity of the ice edge, it decays exponentially further into the ice. This is reflected in the concentration field (Figure 7(e)), which shows that the ice is much more compact at the ice edge. Note that the WRS is not varying significantly in the $y$ direction, showing that the boundary conditions used for the waves at the coast are not having too much influence. Also note that the pack and the MIZ, as shown in the $D_{\max}$ field (Figure 7(d)), are separated by quite a sharp boundary. This has been preserved by doing breaking on the mesh in parallel to the breaking on the grid, as opposed to simply interpolating $D_{\max}$ back to the mesh after doing breaking on the grid. Figure 8 shows the same plot as Figure 7(d), but with this latter, more naive, method of coupling. The sharp MIZ-pack boundary has now become extremely diffuse compared to the former scheme.

Figure 9 tests the sensitivity of the ice edge motion to the rheological parameters $C$ and $\tau_0^L$ when the ice is subjected to steady waves of varying heights (and periods). In Figure 9(a), the damage is set to 0.9999 everywhere the ice is broken by the waves, while in Figure 9(b) the damage and cohesion are unchanged by ice breakage due to waves. Consequently in Figure 9(a) for higher concentrations the internal stress is mainly coming from the ice pressure $P$, while in Figure 9(b) $\boldsymbol{\sigma}$ also plays a role since it is not damaged.

There is a strong response to the compactness factor, $C$, which is used in the neXtSIM model to determine how high the concentration needs to be to increase the effective elastic stiffness and the resistance to ridging to their maximum values. In Figure 9(a), for this initial value of concentration (70%), lowering $C$ by 10 roughly reduces the ice movement by half. Comparing Figure 9(b) to Figure 9(a), if $C = 40$, $\boldsymbol{\sigma}$ makes a difference of between 8–15 km; if it drops to 30, the ice edge movement is approximately reduced by half; if it drops even further to 20, then the ice edge no longer moves at all.

However, the large-scale cohesion makes little difference in these simulations where the ice is not failing. Part of the reason for this is that the wave radiation stress is a compressive stress, so the stresses need to be larger to move outside the Mohr-Coulomb envelope than if they were tensile or shear stresses (see Figure 2: the tensile and shear stresses are near the points of the triangles, while compressive stresses are near their bases).

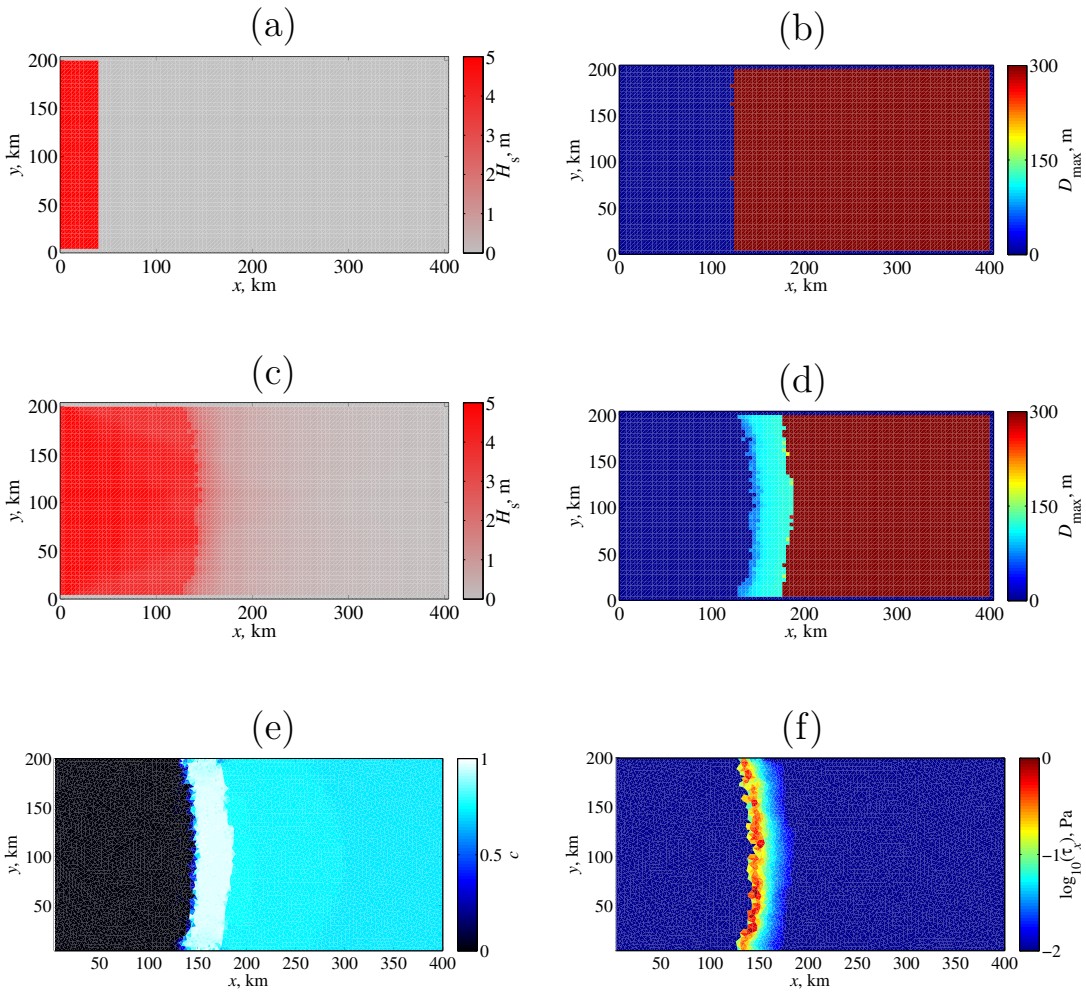

**Figure 7.** Waves breaking ice in an idealized experiment (the right hand, upper, lower lines of grid cells correspond to land). The wave model, based on (Williams et al., 2013a), is coupled to the neXtSIM sea-ice model. The figure shows results after 48 h of steady pushing by a Pierson-Moskowitz wind wave spectrum with significant wave height $H_s = 5\,\text{m}$ (so the peak period $T_p = 11.2\,\text{s}$), that is arriving from the left. It initially occupies the strip shown in (a) then travels to the right, with some directional spreading; the final wave height is shown in (c). (b,d): initial, final maximum floe size (respectively); (e,f): final sea-ice concentration and $x$-component of the wave radiation stress (respectively). The ice has initial conditions (constant where there is ice—see (b) for the initial ice mask): $c = 0.7$, $h = 1\,\text{m}$, $D_{\text{max}} = 300\,\text{m}$, and $d = 0$. Also C=40, $\tau_0^L = 4\,\text{kPa}$, $\tau_0^S = 629\,\text{kPa}$, and $d$ is increased to $d_{\text{break}} = 0.9999$ if the ice is broken.

Some of the runs from Figure 9 (those with $C = 40$ and $\tau_0^L = 4\,\text{kPa}$) were repeated with swell waves (of a single frequency and direction), with amplitude of 3 m and periods ranging from 10–14 s (recalling that the maximum WRS dropped with wave period — Figure 5(b)). These were not able to produce any movement of the ice edge though. Therefore, the main influence of

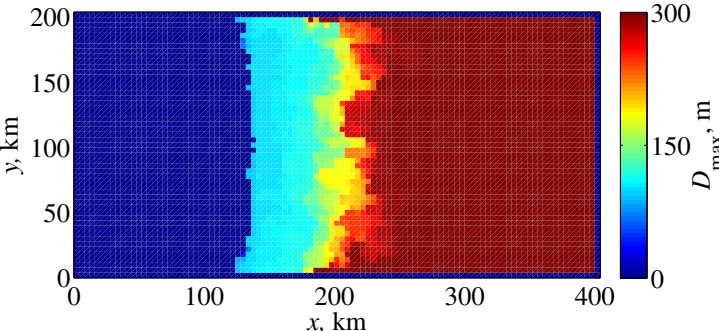

**Figure 8.** Same figure as 7b), but where $N_{\text{floes}}$ is simply interpolated from the neXtSIM mesh onto the WIM grid and then back again after each coupling time-step. Note the boundary between the pack ice and the MIZ has diffused over a large number of grid cells, whereas it has remained much sharper when $N_{\text{floes}}$ is calculated directly on the neXtSIM mesh.

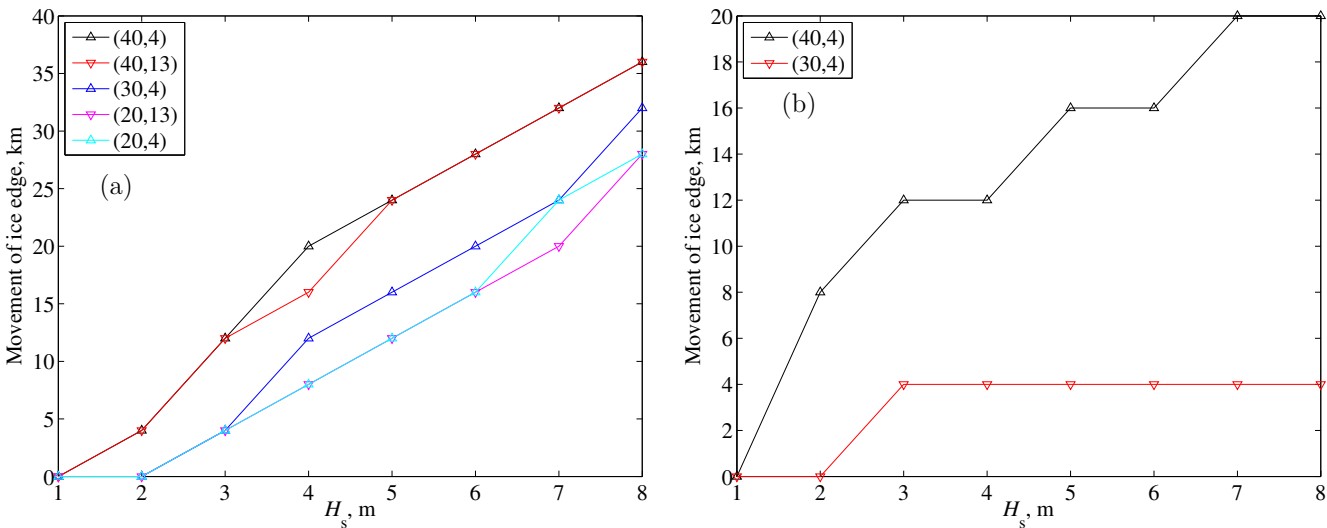

**Figure 9.** Maximum movement of ice edge over 2 days for different pairs $(C, \tau_0^{\text{L}})$ of the compactness factor and the large scale cohesion (in kPa). Initial concentration is 0.7, initial thickness is 1 m. Wave forcing is from Pierson-Moskowitz spectra. (a) Damage is set to $d_{\text{break}} = 0.9999$ if the ice is broken. (b) Damage is unchanged if the ice is broken.

swell will be due to their changing of the dynamical and thermodynamical properties of the ice through the ice break-up. As can be seen from Figures 5–6, they are attenuated less and so they can produce break-up further into the ice than wind waves.

Figure 10 shows the combined effects of wind and waves on the concentration ($c$) and the effective thickness ($ch$). For reference Figures 10(a,b) have only waves (5-m waves following a Pierson-Moskowitz spectrum) and no wind (Figure 10(a) is the same as Figure 7(e)), while Figures 10(c,d) have no waves, but only a $15\,\mathrm{m\,s^{-1}}$ wind from the left (as in Figure 1). This wind speed is consistent with the wind wave spectrum in Figures 10(a,b). Figures 10(e-h) have both 5-m waves and $15\,\mathrm{m\,s^{-1}}$ wind.

All figures with wind Figure 10(c–h) exhibit similar ice edge locations, and all show thickening at the far right "coastline", concentrated in thin "ridges". The area over which the ridging is concentrated also seems similar for all the runs. However, while the pattern of thickening between the three runs seems quite different, perturbations to certain parameters in the run with the R1 modification to the EB rheology (Figures 10(g,h)), such as $d_{\mathrm{break}}$ (0.99 and 0.999 were tried), or the minimum concentration of ice required to cause attenuation (0 or 5% were tried), produce similar degrees of differences. Therefore we

conclude the actual ridging patterns are not significant in themselves. The main differences therefore between the R1 run and the other two are therefore in the concentrations at the ice edge (the actual thickness, $h$, which is not plotted, is constant near the edge). In this run, when the damage is increased if ice breakage occurs, the ice is noticeably more concentrated in a region approximately corresponding to the MIZ. Additionally, the ice edge is more diffuse, possibly due to some feedback effect where if the ice begins to become less concentrated at the ice edge, the attenuation reduces and therefore so does the wave

radiation stress, and then moves more slowly compared to the more concentrated ice which will experience a higher radiation stress — an effect enhanced by the high degree of damage which keeps the more compressed ice quite mobile (as opposed to the run where the rheology is not modified).

Figure 11(a) quantifies the results of Figure 10 with respect to the ice edge location, as well as varying the wind speed. As can be seen from the figure, the waves only increase the movement by $4\,\mathrm{km}$ (no damage in the MIZ due to breakage) or $8\,\mathrm{km}$

(damage is $d_{\mathrm{break}} = 0.9999$ in the MIZ when the ice is broken). That is, the effect of the WRS on the ice edge position is almost completely dominated by the wind stress. When the initial concentration was increased to 95%, the difference was even less (0–4 km), as then the stress and ice pressure $P$ increased due to their $\mathrm{e}^{-C(1-c)}$ factors becoming closer to 1.

To repeat what we have seen in Figure 10, when the ice was subjected to on-ice winds in addition to waves, the main effect of linking the damage to the break-up due to waves was that the MIZ region became more highly compressed than the ice

immediately further in. In Figure 11(b), we see the effects of off-ice winds on ice preconditioned by swell waves. For the wind speed used in the figure shown ($2\,\mathrm{m\,s^{-1}}$), the wind stress is not able to move the pack ice at all, but the MIZ, which is about $60\,\mathrm{km}$ wide and has damage $d_{\mathrm{break}} = 0.9999$, has started to detach from the pack. The ice edge has moved about $15\,\mathrm{km}$ to the left in the centre of the domain, with less movement at the coasts since there is still some friction there (due to the condition of no slip applied at the top and bottom boundaries).

## 6   Conclusions and discussions

In this paper, we have investigated the impact of the WRS on sea-ice state and drift in an idealised domain. While this stress can be quite large ($\sim$.1–1 Pa), depending on the wave conditions, it is extremely localised — decaying exponentially away from the ice edge. Probably as a consequence of this localisation, overall we found its effects on ice edge location were quite

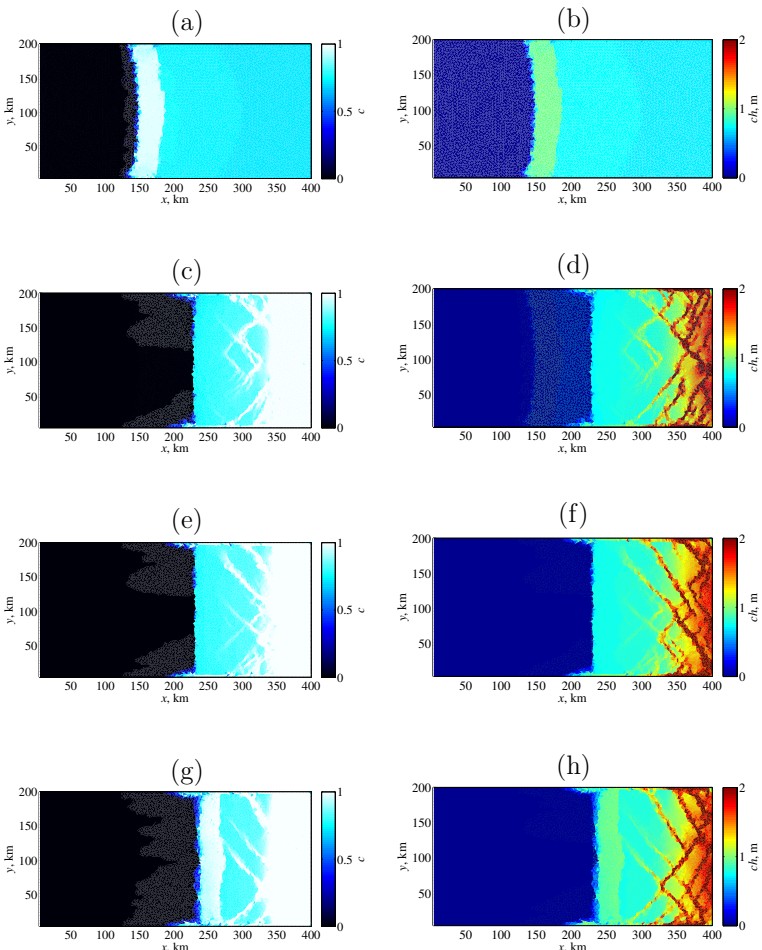

**Figure 10.** (a,b) Concentration ($c$) and effective thickness ($ch$) after the same experiment as Figure 7. (c,d) Same as Figure 1(c,d): wind forcing only. Figures (e–h) show results when steady waves (with $H_{\mathrm{s}} = 5\,\mathrm{m}$, $T_{\mathrm{p}} = 11.2\,\mathrm{s}$, from the left) are applied in addition to the wind forcing. Initial ice conditions are the same as in Figure 7. In (e,f) the ice rheology is not affected by the ice breakage, but in (g,h) damage is set to $d_{\mathrm{break}} = 0.9999$. The large-scale cohesion is $\tau_0^{\mathrm{L}} = 4\,\mathrm{kPa}$, $C = 40$, and the small-scale cohesion is $\tau_0^{\mathrm{S}} = 629\,\mathrm{kPa}$.

modest, with the most noticeable effects being seen when a wind wave spectrum was applied steadily to the ice in the absence of wind. Then, depending on the initial concentration, the rheological parameters used and the response to the ice breakage by waves, the radiation stress could produce a movement of the ice edge of between 0–36 km over two days. However, this experiment is more hypothetical since wind waves are by definition associated with wind. Indeed, in the presence of wind, the wind stress dominated the WRS with almost no difference in ice edge position between experiments with and without waves. There were differences in ridging patterns in the presence of waves but these were probably not significant. However,

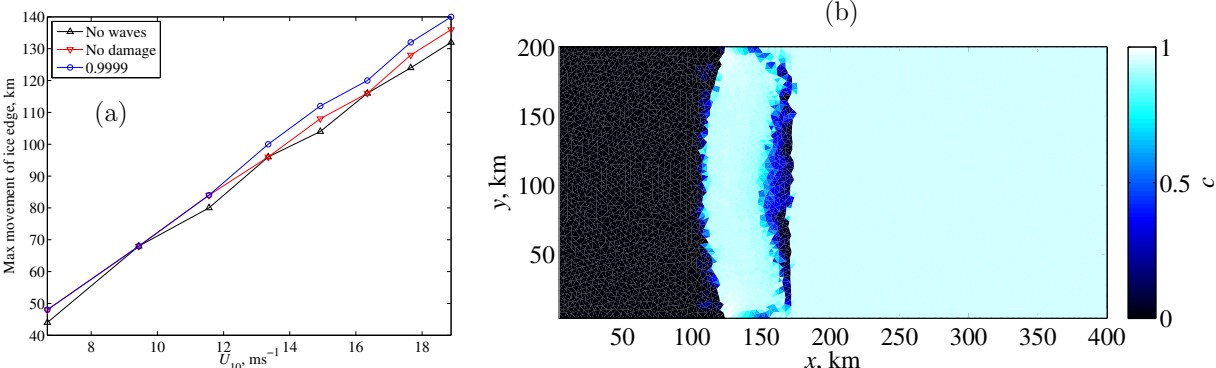

**Figure 11.** (a) shows the maximum movement of the ice edge as a function of wind speed. The different curves show the response to wind forcing only ("no waves"), wind and waves without changing the EB rheology in the MIZ ("no damage"), and wind and waves where the damage is set to $d_{\text{break}} = 0.9999$ in the MIZ ("0.9999"). (b) The effect of swell preconditioning on the response to off-ice winds. Initial conditions: swell of 12 s period and height 3 m is sent into the ice for 24 h, where the ice has constant concentration of 95% and thickness 1 m, breaking the ice for about 50 km. The damage is set to $d_{\text{break}} = 0.9999$ where the ice is broken. Spatially and temporally constant off-ice wind forcing is then applied for a further 48 h, at a speed of $2\,\text{m}\,\text{s}^{-1}$. The large-scale cohesion is 13 kPa, and $C = 40$.

when we modified the damage parameter after ice breakage, additional compression was observed in the MIZ after the ice was broken. Consequently, it seems that the WRS has a very limited effect in general, although it could be a very efficient process to precondition the ice cover and its mechanical properties via the formation of a MIZ area filled with highly damaged ice.

Having said this however, there are many uncertainties regarding the WRS, and we have certainly not included all of its
potential effects, especially since the wave and ice models are not coupled to the ocean yet. For example, the attenuation models are still uncertain (they determine the WRS), and how the partitioning of the WRS between the ice and the ocean should be done is also unknown. On the face of it, if less of the WRS is applied to the ice, it should have even less effect than we find in our current paper. However, perhaps it could then produce similar effects to those discussed and reported by Suzuki and Fox-Kemper (2016) and Suzuki et al. (2016) in relation to overturning circulation produced by the Stokes shear force and
thereby change the currents and heat fluxes acting on the ice.

We also highlighted the problem of numerical diffusion of $N_{\text{floes}}$ due to it being modified by both neXtSIM and the WIM, and therefore having to be communicated in both directions. We presented a solution to this problem, where $N_{\text{floes}}$ was calculated on the neXtSIM mesh each WIM time step, after interpolating smoother wave fields. While not unfeasible, this is somewhat costly and we will continue to look for alternative solutions.
As touched on in the discussion of the WRS above, we also introduced a simple MIZ rheology by increasing the damage where ice was broken, effectively putting the MIZ into free drift, with the addition of the ice pressure which resists compression.

Under compressive wind forcing this led to increased compression in the MIZ relative to the pack ice in its vicinity. This modification also influenced the ice flow when off-ice winds were applied to ice that had previously been broken by swell waves. At lower wind speeds, the MIZ was able to be move relatively freely with the wind, while the pack was still stationary. These effects would undoubtedly be reduced in magnitude were a rheology that represented true granular flow to be used, but could still occur. However, it is difficult to know for certain without the existence of such a rheology. Direct numerical simulations such as those done by Herman (2016) could possibly reproduce some of the effects observed here. Similarly, the granular temperature model of Feltham (2005) could be tried, although this would be limited to flow regimes where large force networks are not expected to be present.

So far we have also restricted ourselves to a simple idealised domain, and with very idealised forcings. Work to set up the current model in a pan-Arctic domain is ongoing, and perhaps studies with forcings with more realistic temporal and spatial variability could find the WRS will have more impact. In addition, the study of Horvat et al. (2016) suggests that including the thermodynamic effects of ice breakage by waves could be important. We are also currently implementing the more conservative lateral melting model of Steele (1992) in our model to include this effect to some extent. With simulations using a WIM coupled to a stand-alone version of CICE-E, which contains the model of Steele (1992), Bennetts et al. (2017) found that the concentration in the vicinity of the Antarctic ice edge could drop by a modest amount (of the order of 10%) in the summer. However, this could also change with coupling to an ocean model, as well as if a different parameterisation that reflects the increased lateral melting of larger floes were used.

## 7  Code availability

This code is not publicly available.

## 8  Data availability

This data is not publicly available.

*Author contributions.*  The paper writing and implementation of the coupling between the WIM and neXtSIM was lead by TW, with formative discussions from PR and SB guiding the progression of the writing. PR also helped with the writing itself, and in addition SB helped implement the coupling.

*Competing interests.*  N/A

*Disclaimer.*  N/A

*Acknowledgements.* This work was primarily supported by the neXtWIM project (Norwegian Research Council grant no 244001). Earlier WIM code development was also supported by the SWARP project (EU-FP7 project 607476) and ONR Global project N62909-14-1-N010. We were also helped by discussions with Einar Ólason and Aleksey Marchenko. Finally we would like to thank our reviewers and editor for their helpful comments.

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
