# Peer review of "Wave-ice interactions in the neXtSIM sea-ice model"

_The Cryosphere, 2017_

## Referee Comment (RC1) · Anonymous Referee #1 · 4 Apr 2017

In their manuscript "Wave-ice interactions in the neXtSIM sea-ice model", T. Williams and colleagues present a coupled wave–sea ice model, based on their neXt-generation Sea Ice Model (neXtSIM). The authors describe in detail the treatment of ice–wave interactions in the coupled model, including sea ice breaking by waves, wave attenuation by ice and calculation of the wave radiation stress (WRS). The model performance is illustrated with results of idealized simulations, in which waves approach sea ice with initially uniform properties, in situations with and without wind.

The manuscript is devoted to a subject which is extremely important for the performance of numerical sea ice models on both short (synoptic) and long (climate) time scales. As the authors point out in the introduction, recent climate change and the associated negative trends in sea ice extent, thickness and strength have produced conditions in which sea ice interactions with waves are becoming more and more important over larger and larger areas. However, our limited understanding of many aspects of these processes is a serious limitation for development of parameterizations of ice–wave interactions suitable for numerical sea ice models. In my opinion, the proposed manuscript is an important contribution to the subject, even though many solutions used in the coupled model and assumptions underlying it are oversimplified (or maybe even wrong). As the authors correctly remark, these simplifying assumptions are a direct result of the lack of observational data and/or theoretical understanding available. In this respect, the most important contribution of the manuscript is that it develops a framework in which future developments can be integrated, as new data and insights become available. In other words, in spite of some clear limitations of the solutions presented, I find this contribution very valuable, as it paves the way for further development.

My comments to the manuscript are listed below. Because addressing them does not require any substantial changes/additions to the content of the paper, my recommendation to the Editors is "minor revision".

1. I have problems with understanding the concept of the damage parameter d in relation to ice concentration c and maximum floe size Dmax. Equation (9c) states that (without thermodynamic effects) d can change (or, more precisely, increase) only if stress falls outside the prescribed envelope. That is, d is not in any direct way related to other quantities characterizing the ice (although, I suppose, in longer simulations the model would by itself evolve into a state in which d, c and Dmax are related). This makes d quite mysterious to me. For example, how should one imagine ice with c = 70% and d = 0, which is used as an initial condition in the simulations? We have relatively dispersed ice, with 30% open water, floes with power-law size distribution (which makes sense only if the number of floes per grid cell is large), but the ice is "undamaged" – how realistic is that? What motivated this choice of initial conditions and how does it influence the results? A more general question: What does "damaged" and "undamaged" mean in physical terms? The authors should include some

discussion/explanation before they proceed to describing details of their model, otherwise some parts of it seem rather obscure. d, c and Dmax are quantities describing the state of the ice cover, and by being shaped by wave–ice interactions they act as a signature of those interactions – so sufficient space should be given to relationships between them. Are all combinations of d, c and Dmax realistic? If not, does the model allow for those combinations or are there some mechanisms that relate one variable to the other?

2. The authors do not describe how Dmax is modified if the breaking criteria (section 3.4) are fulfilled. I know this information can be found in previous papers, but it should be given here for completeness (presumably in section 3.3, together with the description of FSD).

Some minor, technical comments:

1. I'd suggest to replace the word "movement" with "displacement" (or something similar) in the context of the changing position of the ice edge. Especially in the abstract, it is not clear what the sentence ". . . wind waves can produce noticeable movement in loose ice" really means, as no reference to ice edge is made. It wrongly suggests that some analysis of ice motion is made in the paper.

2. The authors should check if all symbols are explained in the text. In most cases (e.g., wave number and amplitude in section 3.4.1) it is obvious what the symbols mean, but still, they should be defined.

3. In the list of references, some papers have missing volume/page numbers, e.g., Meylan et al. 2014 or Rabatel et al. 2015.

---

## Referee Comment (RC2) · Anonymous Referee #2 · 29 May 2017

1. The Coulomb-Mohr criterion can be formulated in terms of principal stresses and in terms of normal and shear stresses (sigma_n, tau) applied to infinitesimal inner surface crossing a material point. The criterion sets that material fails when the stress belongs to the failure envelope. On the plane (sigma_n, tau) the interpretation of the criterion is based on the Mohr circle conception. Material fails when Mohr circle touches the failure envelope. In case of bending deformations the stress state is uniaxial and the first/third principal directions coincides with the horizontal plane. The second principal stress is zero, and the second principal direction is vertical. Therefore the failure occurs due to the tension near the plate surface or plate bottom. Formulas (18f,g) interpret sigma_N and tau as pressure and maximal shear stress. I don't think that it is correct interpretation of the Coulomb-Mohr criterion.

2.The ice failure in continual sea ice models is not similar to ice failure in flexural

strength tests. In the last case the ice is broken by vertical crack due to the bending. It is observed in all tests. In the first case there is damage accumulation, but ice is still represented by continuum. I don't think that it is a good idea to join criterions for large and small scale failure processes even if they look similar. From the other hand in-plane stresses calculated from sea ice dynamic model may influence banding failure by waves since in-plane compression reduces tension caused by the bending. I am not sure that it is important for real application to MIZ problems because there is no strong compression.

Probably above formulated comments will not have significant influence on the results of numerical simulations. In that case I recommend minor revision with improving of the formulations for the failure criterions and correction of Fig. 1.

―――――――――――――――――――――

---

## Author Response (AR1)

**RESPONSE TO ANONYMOUS REVIEWER 1**

We thank the reviewer for their comments.

**1. MAJOR COMMENTS**

(1) *I have problems with understanding the concept of the damage parameter $d$ in relation to ice concentration $c$ and maximum floe size $D_{max}$. Equation (9c) states that (without thermodynamic effects) $d$ can change (or, more precisely, increase) only if stress falls outside the prescribed envelope. That is, $d$ is not in any direct way related to other quantities characterizing the ice (although, I suppose, in longer simulations the model would by itself evolve into a state in which $d$, $c$ and $D_{max}$ are related). This makes $d$ quite mysterious to me. For example, how should one imagine ice with $c = 70\%$ and $d = 0$, which is used as an initial condition in the simulations? We have relatively dispersed ice, with 30% open water, floes with power-law size distribution (which makes sense only if the number of floes per grid cell is large), but the ice is "undamaged" — how realistic is that? What motivated this choice of initial conditions and how does it influence the results? A more general question: What does "damaged" and "undamaged" mean in physical terms? The authors should include some discussion/explanation before they proceed to describing details of their model, otherwise some parts of it seem rather obscure. $d$, $c$ and $D_{max}$ are quantities describing the state of the ice cover, and by being shaped by wave-ice interactions they act as a signature of those interactions — so sufficient space should be given to relationships between them. Are all combinations of $d$, $c$ and $D_{max}$ realistic? If not, does the model allow for those combinations or are there some mechanisms that relate one variable to the other?*

  (a) With regard to the unfamiliarity of the reviewer (and probably other readers) with the damage variable $d$ we have added a new section (now §2.3) with an example simulation showing its role, which is usually to produce localised damage and linear kinematic features.

  (b)    (i) With regard to the initial conditions, one of the main results of the paper was that the WRS had relatively little effect in most situations, but we still wanted to give a demonstration of situations where it did do something, even if those situations were relatively rare. This led to the choice of $c = 0.7$ initially, because for higher concentrations ($\gtrsim 0.9$) the internal stress could balance the WRS without failing (ie the WRS did nothing for higher concentrations) — see the formulae for $Y_*(c, d)$ and $P$.

     (ii) Not being an observable variable, $d$ is difficult to initialise so we initialise it to zero usually and let it evolve according to the different forcings, given the

other variables which are observable. In general, there is no explicit relationship between it and $c$. However, we can make some generalisations about it — eg. it is only increased if $c$ is high, when the internal stress becomes large enough to cause the ice to fail. Thus initially having $c = 0.7$ and $d = 0$ is not an unrealistic combination.

(iii) We assumed the ice was initially unbroken — ie we initialised $D_{\max}$ to its default "large" value of $300\,\mathrm{m}$ for simplicity mainly, but we note that it is not incompatible with the initial value of $c$, since in summer it is possible to have large floes with large gaps between them (smaller floes melt faster). In general, $d$ and $D_{\max}$ are not related, but in some simulations we employed a rudimentary MIZ "rheology" where we set the damage to a high value when it was broken — this lowers the effective elastic stiffness $Y_*$ to near zero, putting the ice nearly in free drift, although $P$ provides some resistance to compaction.

(iv) We have added some extra descriptions/justifications of our initial conditions.

(2) *The authors do not describe how $D_{max}$ is modified if the breaking criteria (section 3.4) are fulfilled. I know this information can be found in previous papers, but it should be given here for completeness (presumably in section 3.3, together with the description of FSD).* A new section with this information is added now (section 3.4.4)

**2. Minor comments**

(1) *I'd suggest to replace the word "movement" with "displacement" (or something similar) in the context of the changing position of the ice edge. Especially in the abstract, it is not clear what the sentence "...wind waves can produce noticeable movement in loose ice" really means, as no reference to ice edge is made. It wrongly suggests that some analysis of ice motion is made in the paper* We have clarified that we only looked at the displacement of the ice edge in the abstract and elsewhere.

(2) *The authors should check if all symbols are explained in the text. In most cases (e.g., wave number and amplitude in section 3.4.1) it is obvious what the symbols mean, but still, they should be defined.* We have checked that all symbols are defined now.

(3) *In the list of references, some papers have missing volume/page numbers, e.g., Meylan et al. 2014 or Rabatel et al. 2015.* These have been fixed.

**RESPONSE TO ANONYMOUS REVIEWER 2**

We thank the reviewer for their comments.

(1) *Material fails when Mohr circle touches the failure envelope...Formulas (18f,g) interpret $\sigma_N$ and $\tau$ as pressure and maximal shear stress. I don't think that it is correct interpretation of the Coulomb-Mohr criterion.* The reviewer is correct in general, but in this situation where $\sigma_{12} = 0$ it is fine to use $\sigma_{11}$ and $\sigma_{22}$. However, we have clarified this point by defining the principal stresses explicitly.

(2) *The ice failure in continual sea ice models is not similar to ice failure in flexural strength tests. In the last case the ice is broken by vertical crack due to the bending. It is observed in all tests. In the first case there is damage accumulation, but ice is still represented by continuum. I don't think that it is a good idea to join criterions for large and small scale failure processes even if they look similar.* In fact Mohr-Coulomb failure has been observed on many scales (eg. Schulson *et al.*, 2006), with the size of the failure envelope (the cohesion) depending on the defect size, which corresponds to the scale considered. Thus the small scale cohesion was of the order $10^5$–$10^6$ Pa, while the large scale cohesion was of the order $10^3$–$10^4$ Pa. Note however that we do not have a single merged failure criterion for both the floe and the mesh scales, but in fact have two independant criteria. What may have led to some confusion, is that in some of the experiments, we applied a rudimentary MIZ dynamical model by setting the damage parameter $d$ to a high value so that the ice was effectively in free drift (unless it was converging, when the ice pressure $P$ was activated). This $d$ parameter was also the same one changed if the large scale stresses left the large-scale failure envelope. We clarify this in the text.

$$\sigma_{22} = \frac{\nu Y}{1 - \nu^2},$$

while our old equation (18e) had a minus sign in front of this. This had the effect that the definition of the slope of the $\sigma_N - \tau$ curve for a plane wave changes to $\alpha = (1 - \nu)/(1 + \nu)$, and the values of $\sigma_{11}$ where this curve intersects the Mohr-Coulomb envelope change to

$$\sigma_{11}^+ = \frac{2\sigma_N^+}{1 + \nu} = \frac{2\tau_0^{\mathrm{S}}}{(1 + \nu)(\alpha + \mu)} = \frac{2\tau_0^{\mathrm{S}}}{1 - \nu + \mu(1 + \nu)} \equiv \gamma\tau_0^{\mathrm{S}},$$

$$\sigma_{11}^- = \frac{2\sigma_N^-}{1 + \nu} = -\frac{2\tau_0^{\mathrm{S}}}{(1 + \nu)(\alpha - \mu)} = -\frac{2\tau_0^{\mathrm{S}}}{1 - \nu - \mu(1 + \nu)},$$

while the old formulae were

$$\sigma_{11,\mathrm{bad}}^+ = \frac{2\tau_0^{\mathrm{S}}}{1 + \nu + \mu(1 - \nu)} \equiv \gamma_{\mathrm{bad}}\tau_0^{\mathrm{S}} = \gamma\left(\frac{\gamma_{\mathrm{bad}}}{\gamma}\tau_0^{\mathrm{S}}\right),$$

$$\sigma_{11,\mathrm{bad}}^- = -\frac{2\tau_0^{\mathrm{S}}}{1 + \nu - \mu(1 - \nu)}.$$

That is, the mapping between cohesion $\tau_0^{\mathrm{S}}$ and the value of $\sigma_{11}$ at failure was incorrect. The effect of this error is then that the value we thought we were using for the small scale cohesion $\tau_0^{\mathrm{S}}$ was actually $(f_{\mathrm{bad}}\tau_0^{\mathrm{S}})$, where the correction factor is

$$f_{\mathrm{bad}} = \frac{\gamma_{\mathrm{bad}}}{\gamma} \approx 0.899.$$

In practice, we have amended this error by changing the values of $\tau_0^{\mathrm{S}}$ in the legends, figure captions and discussion of figures to the correct values of $\tau_0^{\mathrm{S}}$. While these new values are slightly less logical, they still approximately cover the intended range of cohesions (.25–1 MPa).

(2) We have clarified the assumptions about the Mohr-Coulomb criterion in a random sea (see the new section 3.4.2). In particular, the situation becomes more complicated when multiple directions are present — for now we simplify matters by assuming unidirectional waves (during the breaking process only), but this should be revisited in future work.

(3) The discussion of the correspondence between the flexural strength and the cohesion for an Euler-Bernoulli beam was corrected (see the new section 3.4.3). This didn't impact the numerical results however.

(4) A new section illustrating the role of the damage variable was added (new section 2.3).

[revised manuscript text omitted]

E_{\text{K}} &= \rho_{\text{w}} \left\langle \int_{z_{\text{bot}}}^{\eta} (u_{\text{w}}^2 + v_{\text{w}}^2)\,\mathrm{d}z \right\rangle \approx \rho_{\text{w}} \int_{z_{\text{bot}}}^{0} \langle u_{\text{w}}^2 + v_{\text{w}}^2 \rangle \,\mathrm{d}z \\
&= \frac{\rho_{\text{w}}\omega^2 a^2}{4k} \rho_{\text{w}}\omega^2 \frac{A^2}{4k} \cosh(kz_{\text{bot}} kZ) = \rho_{\text{w}}g \frac{a^2}{4} \cdot \frac{A^2}{4},
\end{aligned}
\tag{33}
$$

where $u_{\text{w}}$ and $v_{\text{w}}$ are the horizontal and vertical wave orbital velocities, and $Z$ is the water depth. In a conservative system, the mean potential energy and the mean kinetic energy are equal, so the mean energy density is simply

$$
E_{\text{tot}} = 2E_{\text{K}} = \rho_{\text{w}}g \frac{a^2}{2} \frac{A^2}{2} = \rho_{\text{w}}g \langle \eta^2 \rangle.
\tag{34}
$$

The mean momentum per unit area is:

$$
\begin{aligned}
\mathbf{M} &= \left\langle \int_{z_{\text{bot}} - Z}^{\eta} (u_{\text{w}}, v_{\text{w}})\,\mathrm{d}z \right\rangle = -\rho_{\text{w}} \langle \Phi_{z=\eta} \nabla\eta \rangle \approx -\rho_{\text{w}} \langle \Phi_{z=0} \nabla\eta \rangle \\
&= \rho_{\text{w}}g \frac{ka^2}{2\omega} \frac{kA^2}{2\omega} (\cos\theta, \sin\theta) = \frac{E_{\text{tot}}}{c_{\text{p}}} (\cos\theta, \sin\theta),
\end{aligned}
\tag{35}
$$

where $c_{\text{p}} = \omega/k$ is the phase velocity.

When we consider a complete wave spectrum, then

$$
\mathbf{M} = \rho_{\text{w}}g \int_{0}^{\infty} \int_{0}^{2\pi} \frac{E(\mathbf{x};\omega,\theta)}{c_{\text{p}}} (\cos\theta, \sin\theta)\,\mathrm{d}\theta\mathrm{d}\omega,
\tag{36}
$$

and its flux is

$$
\begin{aligned}
D_t\mathbf{M} &= \rho_{\text{w}}g \int_{0}^{\infty} \int_{0}^{2\pi} \frac{D_t E(\mathbf{x};\omega,\theta)}{c_{\text{p}}} (\cos\theta, \sin\theta)\,\mathrm{d}\theta\mathrm{d}\omega \\
&= \rho_{\text{w}}g \int_{0}^{\infty} \int_{0}^{2\pi} \frac{S_{\text{ice}}(\mathbf{x};\omega,\theta)}{c_{\text{p}}} (\cos\theta, \sin\theta)\,\mathrm{d}\theta\mathrm{d}\omega.
\end{aligned}
\tag{37}
$$

This quantity can then be transferred to the ice, ocean and atmosphere, according to the different attenuation mechanisms, i.e.

$$
-D_t\mathbf{M} = \tau_{\text{w,i}} + \tau_{\text{w,o}} + \tau_{\text{w,a}}.
\tag{38}
$$

For this study we assume that all the momentum goes to the ice — i.e. $\tau_{\text{w,o}} = \tau_{\text{w,a}} = 0$.

[revised manuscript text omitted]

15 ## 5   Results

**5.1   Note on wave and wind forcing**

In our results section we will partly use incident wind wave spectra based on the Bretschneider spectrum:

$$E_{\text{B}}(\omega; H_{\text{s}}, \omega_{\text{p}}) = \frac{5 H_{\text{s}}^2 \omega_{\text{p}}^4}{16 \omega^5} \mathrm{e}^{-(5\omega_{\text{p}}^4)/(4\omega^4)}, \tag{39}$$

where $H_{\text{s}}$ is the significant wave height, $\omega_{\text{p}} = 2\pi/T_{\text{p}}$, and $T_{\text{p}}$ is the peak period.

20 Since $H_{\text{s}}$ and $T_{\text{p}}$ are not totally independent, to try to make them roughly consistent we will also use a special case of (39), the Pierson-Moskowitz spectrum which was defined as an approximation for fully-developed wind seas:

$$E_{\text{PM}}(\omega; \underline{H_{\text{s}}}, \omega_{\underline{\text{p}0}}) = \frac{a_{\text{PM}} g^2}{\omega^5} \mathrm{e}^{-b_{\text{PM}}(\omega_0/\omega)^4}, \tag{40}$$

where $a_{\text{PM}} = 8.1 \times 10^{-3}$, $b_{\text{PM}} = 0.74$, and $\omega_0 = g/U_{19.5} \approx g/(1.026 U_{10})$. Here $U_{19.5}$ and $U_{10}$ are the wind speeds 19.5 m and 10 m above the sea (respectively) — note that these wind speeds are linked to the incident wave parameters, and we will also try

to keep them consistent when we are presenting coupled WIM-neXtSIM results. The Bretschneider parameters corresponding to the Pierson-Moskowitz parameters are:

$$\omega_{\mathrm{p}} = (4b_{\mathrm{PM}}/5)^{1/4}\omega_0 \approx 0.877\omega_0, \tag{41a}$$

$$H_{\mathrm{s}} = \frac{4g}{\omega_{\mathrm{p}}^2}\sqrt{\frac{a_{\mathrm{PM}}}{5}}. \tag{41b}$$

Our incident wind wave spectra will then combine a Bretschneider frequency spectrum with some directional spreading:

$$E_{\mathrm{inc}}(\omega,\theta;H_{\mathrm{s}},\omega_{\mathrm{p}}) = E_{\mathrm{B}}(\omega;H_{\mathrm{s}},\omega_{\mathrm{p}})D_{\mathrm{inc}}(\theta), \;\; D_{\mathrm{inc}}(\theta) = \frac{2}{\pi}\cos^2\theta \times H(|\theta|-\pi/2), \tag{42}$$

where $H$ is the Heaviside step function. (Note that the mean wave direction is zero, ie to the right in our model domain, which can be seen in Figure 1.) We will also look at so-called swell waves, which are not locally generated, generally quite long (wave period greater than about 10 s or longer), and are monochromatic and mono-directional.:

$$E_{\mathrm{swell}}(\omega,\theta;H_{\mathrm{swell}},\omega_{\mathrm{swell}}) = \frac{1}{8}H_{\mathrm{swell}}^2\delta(\omega-\omega_{\mathrm{swell}})\delta(\theta). \tag{43}$$

**5.2 Sensitivity of MIZ width to Young's modulus and small-scale cohesion**

The purpose of this section is to test sensitivity to the Young's modulus and the small-scale cohesion, not necessarily to decide on "correct" values, which are best determined from future observations. The experiments are similar to those of Williams *et al.* (2013*b*), although the effect of the Young's modulus was not tested in that paper. This is an interesting parameter since increasing it makes the ice less compliant and easier to break (ie. a given wave amplitude produces a higher stress in the ice) — potentially increasing the MIZ width — but this also increases the attenuation, which could potentially reduce the MIZ width. The effect of the small-scale cohesion will play a similar role to the breaking strain in that paper.

The Young's modulus is typically somewhere in the range of 1–10 GPa. Williams *et al.* (2013*a*) argued for values within the interval 5–7 GPa (depending on the brine volume fraction), proposing that the effective elastic modulus, which includes a response to primary, recoverable creep, should cause it to drop somewhat from the relationship of Timco & Weeks (2010). However, Marchenko *et al.* (2013) derived significantly lower values of Young's modulus (about 1.5 GPa) in Svalbard fjord ice. Marchenko *et al.* (2017) also measured lower values in the Barents Sea, ranging between 1–4 GPa, with no obvious dependance on the brine volume. Therefore, we do some tests of the sensitivity of the MIZ width and the maximum WRS to this parameter.

Figure 4 shows the variation of (a) the MIZ width and (b) the maximum WRS with peak period for different values of the Young's modulus. Since increasing the Young's modulus increases the attenuation, the waves lose more momentum and so the maximum radiation stress increases, and this is clearly seen in (b). However, (a) clearly shows that the MIZ width increases with increasing Young's modulus, so its effect on the breaking criterion clearly dominates its effect on the attenuation. The magnitude of the maximum radiation stress is of the order of 0.1–1 Pa, which is comparable to the wind stress from a 10–20 10–15 m s$^{-1}$ winds (if the ice-air drag coefficient is $2\times10^{-3}$ kgm$^{-3}$, this range of wind speeds corresponds to a range in $\tau_{\mathrm{a}}$ of 0.2–0.8 Pa). 
[revised manuscript text omitted]

---

## Author Response (AR2)

Comments to the Author:
Dear Drs. Williams, Rampal and Bouillon,

Thank you for your response to the reviewers, and thank you to the reviewers for their time and consideration.

Please proof read the paper. Some specific edits to consider are given below.
*This has now been done.*

1. Abstract line 8: "too little to induce a very large WRS". This is vague wording, if you can quantify or clarify do.
*This sentence has been rephrased.*

2. I suggest you be consistent in your hypenation of sea ice. When it is a compound adjective (as in the title of the paper), you may hypenate. If you do, do so through out the manuscript. e.g. pg 2 line 23 is missing a hypen. Note, it is up to you whether you decide to hyphenate sea ice or not. I do not see a journal policy on this. Just be consistent.
*We have now used "sea-ice" throughout when it is a compound adjective.*

Figure 2 is referenced before Figure 1. You could avoid this by moving the new section 2.3 up in the manuscript. It seams out of place where it is anyway.
*We have swapped sections 2.2 and 2.3.*

Page 12, line 10 (and elsewhere), puctuate e.g..
*We have changed to e.g. throughout.*

Page 13, line 1: There is one to many 'to's in this line. Also ration -> ratio
*Corrected.*

Section 5.1: Should this be in the results section?
*We think it is best here as it is not really anything original theoretically, but only a not on the forcings used in the results section. If you still think it should be moved we could move it to an appendix perhaps?*

Page 17 and elsewhere: Please reference the figures with their number and panel letter. From the instructions for authors:
"The abbreviation "Fig." should be used when it appears in running text and should be followed by a number unless it comes at the beginning of a sentence, e.g.: "The results are depicted in Fig. 5. Figure 9 reveals that...".
It is unclear which figure you are refering to when you just reference the letters. At least this is the case when reading too quickly!
*Figure references corrected, although we haven't abbreviated – should we do this?*

Page 18, line 1: "solid curves in Fig. ?? are created"
*Figure reference added.*

Page 19, line 10: is (42) a missing reference?

*It is an equation - this has now been clarified.*

Sincerely,
Jenny

Non-public comments to the Author:
This article is an interesting contribution to the conversation on how to model wave-ice interaction. There are some aspects of the experiment set-up that are perhaps not realistic in the context of sea ice, and I appreciate that you point these out.

Consider thanking the reviewers in the acknowledgements if you found their input valuable.
*Acknowledgements updated.*

---

## Author Response (AR3)

**Reply to reviewer 2:**

**"Comments to failure criterions described in the paper"**

Figure 1. Ice failure envelopes on the plane of principal stresses (a) and Coulomb-Morh plane (b). SFE – Schulson's failure envelope, WFE – Williams failure envelope.

Failure envelopes on the planes of principal stresses (PPS) ( $\sigma_1$ ,  $\sigma_2$ ) and plane of normal and shear stresses (Coulomb-Morh plane or PCM) ( $\sigma_n$ ,  $\tau_n$ ) are considered. Mapping of failure envelopes from PPS to PCM is only possible for the points where the tangent lines are inclined to the line C45 (Fig. 1a) under the angle less than 45°. Points C1,2 and D1,2 are mapped in the points OC and OD (Fig. 2). WFE interpolates SFE linearly in the region where principal stresses are positive. It has a shape similar to shown in Fig. 1a by lines WFE. If slope angle of the WFE to the line C45 are less than 45° then the WFE is mapped on PCN in two lines denoted as WFE in Fig. 1b. SFE assumes that the ratio of compressive strength  $\sigma_c$  to tensile strength  $\sigma_t$  is about 5.

Stress state in 2D plane bending is characterized by one nonzero principal stress, while the other principal stress is zero. Absolute value of nonzero principal stress reaches maximum at the plate surface and plate bottom. Since tensile strength is lower compressive strength then the plate fail in places where principal stress is positive. Stress state of a material point is characterized by the point on PPS with coordinates ( $\sigma_1$ ,  $\sigma_2$ ), where  $\sigma_2$ =0 when  $\sigma_1$ >0, and  $\sigma_1$ =0 when  $\sigma_2$ <0. Thus principal stresses inside the plate sit on segments OO1 and OO2 (Fig. 1a). Point O1 has coordinates ( $\sigma_t$ ,0), and point O2 has coordinates ( $\sigma_{\tau}$ ).

Stress state of a material point is characterized by the Mohr circle on PCM. The center of the Mohr circle sits on the axis  $\sigma_n$ , the radius of the Mohr circle equals  $(\sigma_1 - \sigma_2)/2$ , and the Morh circle crosses axis  $\sigma_n$  in points with coordinates  $\sigma_n = \sigma_1$  and  $\sigma_n = \sigma_2$ . Material fails when the Morh circle touches the failure envelope. Since one of the principal stresses in bended plate is always zero then the stress state of the plate points with tensile stresses is performed by a set of Mohr circles Mt shown in Fig. 1b. They touch vertical axis  $\tau_n$  in the origin and extended in the region of positive stresses. These Mohr circles touch the failure envelope when the diameter of the Mohr circle equals tensile strength  $\sigma_t$ . Thus there is no difference in the consideration of failure criterions on PPS and PCM. According to SFE the plate is broken when maximal tensile stress reaches  $\sigma_t$ , and according to SFE and WFE.

**Reply**

We thank the reviewer for their comprehensive discussion on Mohr-Coulomb failure. We have amended sections 2.3 and 3.4.1-3.4.2 so that the failure criterion is applied in the PPS, in the manner of Dansereau et al (2016) and Rampal et al (2016). In fact, the failure criterion was already done in this way inside neXtSIM – it was only the description that was incorrect. The breaking criterion under plane wave forcing needed amending to determine the proper point when the wave stress met the failure envelope. In practice this meant the conversion from small-scale cohesion to breaking stress/strain needed correcting, so in addition to the changes to the theoretical description in 3.4.1-3.4.2, the results section needed changing so that the correct values of small-scale cohesion were stated in captions and figure legends.

Further comments:

- We do indeed use linear interpolation to the Schulson (2006) envelope in the 1st quadrant (as Schulson himself does). Other data (eg Weiss et al, 2007, fig 2) also are quite well fitted by a linear envelope in this quadrant.
- We have extended this linear fit into the other quadrants, which is also consistent with the data of Weiss et al (2007). This is discussed in detail in section 2.3. Similarly, the closing of the envelope for high compressions is discussed briefly. Neither of these have large impact on failure inside the ice rheology, but the extension of the envelope into the 3rd quadrant would make a difference to the wave failure criterion, since it produces tensile failure. However, with the uncertainty in this failure criterion, we don't feel it is worth dwelling on too much. Stress measurements for ice during break-up by waves would be extremely interesting from this point of view but would also be very difficult to obtain.

**References**:**

- 1. Dansereau, V., Weiss, J., Saramito, P., and Lattes, P.: A Maxwell elasto-brittle rheology for sea ice modelling, The Cryosphere, 10, 1339–1359, 2016.
- 2. Rampal, P., Bouillon, S., Ólason, E., and Morlighem, M.: neXtSIM: a new Lagrangian sea ice model, The Cryosphere, 10, 1055–1073, doi:10.5194/tc-10-1055-2016, 2016.
- 3. Schulson, E. M., Fortt, A. L., Iliescu, D., and Renshaw, C. E.: Failure envelope of first-year Arctic sea ice: The role of friction in compressive fracture, Journal of Geophysical Research: Oceans, 111, n/a–n/a, doi:10.1029/2005JC003235, http://dx.doi.org/10.1029/2005JC003235, c11S25, 2006.
- 4. Weiss, J., Schulson, E. M., and Stern, H.: Sea ice rheology from in-situ, satellite and laboratory observations: Fracture and friction, Earth and Planetary Science Letters, 255, 1–8, doi:10.1016/j.epsl.2006.11.033, 2007.

**CHANGE LOG**

All changes can be seen in the difference between the submitted and revised manuscripts (shown below). The most important changes relates to correcting the Mohr-Coulomb envelope for the wave breakage. Note we have changed sign convention to the previous revision, so that tensile stresses are now negative.

With the corrected failure envelope, we have failure when

$$\sigma_1 = \frac{-2\tau_0^{\rm S}/(q-\nu)}{\sqrt{\mu^2 + 1 - \mu}} \equiv \gamma \tau_0^{\rm S},$$

while in the previous revision we had failure when

$$\sigma_1 = \frac{-2\tau_0^{\rm S}}{1 - \nu + \mu(1 + \nu)} \equiv \gamma_{\rm bad} \tau_0^{\rm S} = \gamma \left(\frac{\gamma_{\rm bad}}{\gamma} \tau_0^{\rm S}\right).$$

That is, the mapping between cohesion  $\tau_0^{\rm S}$  and the value of  $\sigma_1 = \sigma_{11}$  at failure was incorrect. The effect of this error is then that the value we thought we were using for the small scale cohesion  $\tau_0^{\rm S}$  was actually  $(f_{\rm bad}\tau_0^{\rm S})$ , where the correction factor is

$$f_{\rm bad} = \frac{\gamma_{\rm bad}}{\gamma} \approx 1.10$$

In practice, we have amended this error by changing the values of  $\tau_0^{\rm S}$  in the legends, figure captions and discussion of figures to the correct values of  $\tau_0^{\rm S}$ .

**Wave-ice interactions in the neXtSIM sea-ice model**

Timothy D. Williams1, Pierre Rampal1, and Sylvain Bouillon1

1Nansen Environmental and Remote Sensing Center, Thormøhlensgate 47, N5006, Bergen, Norway and the Bjerknes Center for Climate Research, Bergen, Norway.

Correspondence to: T. D. Williams (timothy.williams@nersc.no)

**Abstract.** In this paper we describe a waves-in-ice model which calculates ice breakage and the wave radiation stress (WRS) that is coupled to the new sea-ice model neXtSIM, which is based on the Elasto-Brittle (EB) rheology. We highlight some numerical issues involved in the coupling, and investigate the impact of the WRS, and of modifying the EB rheology to lower the stiffness of the ice in the area where the ice has broken up (the marginal ice zone, or MIZ).

- In experiments in the absence of wind, we find that wind waves can produce noticeable movement of the ice edge in loose ice (concentration around 70%) — up to 36 km, depending on the material parameters of the ice that are used, and the dynamical model used for the broken ice. The ice edge position is unaffected by the WRS if the initial concentration is higher ( $\gtrsim 0.9$ ). Swell waves (monochromatic waves with low frequency) do not affect the ice edge location (even for loose ice), as they are attenuated much less than the higher frequency components of a wind wave spectrum, and so consequently produce a much
- 10 lower WRS (by about an order of magnitude at least).

In the presence of wind, we find that the wind stress dominates the WRS, which while large near the ice edge, decays exponentially away from it. This is in contrast to the wind stress which is applied over a much larger ice area. In this case (when wind is present) the dynamical model for the MIZ has more impact than the WRS, although that effect too is relatively modest. When the stiffness in the MIZ is lowered due to ice breakage, we find that on-ice winds produce more compression in the MIZ then in the peak while off ice winds can ease the MIZ to be compared from the peak ice.

[revised manuscript text omitted]
_{\underline{N}2} = (\underbrace{\nu}\sigma_{\underline{1}} + \sigma_{\underline{2}})_{\underline{1}} = (\underline{1}\sigma_{\underline{c}} + \underbrace{\nu})q\sigma_{\underline{11}1}, \tag{21a}$$

$$\underline{\tau\sigma_1} = \underline{(\sigma_1_1^{(\text{tens})} \equiv -\underline{\sigma_2})}_{\underline{q-\nu}} = \underline{(1-\nu)\sigma_{11}} = \alpha\sigma_N, -\frac{(2\tau_0^S)/(q-\nu)}{\sqrt{\mu^2 + 1 - \mu}} \approx -1.13\tau_0^S$$
(21b)

5 where  $\alpha \equiv (1 - \nu)/(1 + \nu)$ . The maximum strains are produced when  $z = \pm h/2$  (at the upper and lower surfaces of the ice), and so for a plane wave-

$$\underbrace{\varepsilon \equiv \max\{\varepsilon_{11}\} = \frac{1}{2}k^2Ah.}_{=}$$

For a wave spectrum, the corresponding quantity to (22)is related to the maximum mean square strain by

$$\frac{\varepsilon^2}{2} \equiv \left\langle \max\{\varepsilon_{11}\}^2 \right\rangle = m_{\varepsilon}, \ m_{\varepsilon} \equiv \frac{\hbar^2}{4} \int_{0}^{\infty} \int_{0}^{2\pi} E(\mathbf{x}, t; \omega, \theta) k^4 \, \mathrm{d}\theta \, \mathrm{d}\omega.$$

10 Here we have assumed that all the wave energy is directed in one direction (which direction is not relevant since we also do not attempt to consider an anisotropic wave medium). We will return to this assumption in the following section. if μ = 0.7, it doesn't meet the lower branch, σ1 = σc + qσ2, if σN ≥ σN.min). Note that here the shape of the tip of the failure envelope makes a difference, since a pure tensile failure criterion would increase the lower limit on σ1 to -σc/q ≈ -1.04τ0S (which would be reached at smaller wave amplitudes). However, given the uncertainty about the failure envelope under pure tension and high
15 compression, and so that our small- and large-scale envelopes have the same shape, we use (11) for wave failure also.

Figure 2(a) plots the failure envelopes for two values of the cohesion. The figure also shows where the line corresponding to the stress state for plane waves,  $\tau = \alpha \sigma_N \sigma_2 = \nu \sigma_1$ , meets these Mohr-Coulomb envelopes . This happens when

$$\sigma_N = \sigma_N^{\pm} = \pm \frac{\tau_0^{\mathbf{S}}}{\alpha \pm \mu};$$

25

the '+' corresponds to tensile failure, while the '-' corresponds to compressive failure. The stress  $\sigma_{11}$  at these points is given 20 by-

$$\underline{\sigma_{11}^{+}} = \frac{2\sigma_{N}^{+}}{1+\nu} = \frac{2\tau_{0}^{S}}{(1+\nu)(\alpha+\mu)} = \frac{2\tau_{0}^{S}}{1-\nu+\mu(1+\nu)} \approx 1.24\tau_{0}^{S},$$

$$\overline{\sigma_{11}^{-}} = \frac{2\sigma_{N}^{-}}{1+\nu} = -\frac{2\tau_{0}^{S}}{(1+\nu)(\alpha-\mu)} = -\frac{2\tau_{0}^{S}}{1-\nu-\mu(1+\nu)} \approx -9.5\tau_{0}^{S}$$

(using  $\mu = 0.7$  (i.e.,  $\nu = 0.3$ ). Therefore the ice will fail under tension first. Note however, that  $\sigma_N \approx 0.8\tau_0^S$  always reaches the upper Coulomb branch ( $\tau = \mu \sigma_N$ ) before it exceeds the maximum tensile strength ( $\sigma_{N,max} \approx 1.2\tau_0^S$ , again using  $\mu = 0.7$ ,  $\nu = 0.3$  when  $\sigma_1 = \sigma_1^{(tens)}$ ).